# A plastid carbohydrate carrier mediates ribose recycling from nucleotide catabolism and glucose export from starch degradation

Luisa Voß [1], Isabel Keller[2], Rebekka Schröder[1], Denise Mehner-Breitfeld[3], André Specht[1], Gerald Dräger[4], Jannis Rinne [5], Jakob Franke [5], Nieves Medina-Escobar[1], Marco Herde [1], Thomas Brüser[3], H. Ekkehard Neuhaus [2] & Claus-Peter Witte [1] ✉

In plants, nucleotide degradation releases ribose in the cytosol. An unidentified transporter then brings the ribose into the plastids for phosphorylation. This process of ribose recycling is particularly prominent in root nodules of soybean (*Glycine max*) and common bean (*Phaseolus vulgaris*) during symbiotic nitrogen fixation. In this biological context, we identified a plastid ribose transporter, which is an ortholog of the putative plastid glucose transporter (pGlcT) of *Arabidopsis thaliana*. We show that Arabidopsis mutants of *At-pGlcT*, but not of the related *At-pGlcT2*, accumulate ribose and fructose constitutively, whereas glucose accumulates only at night. Uridine feeding experiments leading to cytosolic ribose release indicated that *At*-pGlcT transports ribose from the cytosol into the plastids. Uptake assays with complemented *Escherichia coli* sugar transport mutants directly demonstrated that *At*-pGlcT transports ribose, glucose, and fructose. Ribose and fructose accumulation were also observed in CRISPR-induced bean nodule mutants of *Pv-pGlcT*. Additionally, our data show that ribose recycling is important for producing allantoin, a nitrogen fixation product used for nitrogen export from nodules to shoots. We conclude that pGlcT is a plastid facilitator for the import of ribose from nucleotide catabolism, for the export of glucose from nocturnal starch breakdown, and for cytosol-plastid fructose exchange in vivo.

Plants are excellent recyclers and can recover nutrients from metabolites or macromolecules. For example, they can fully degrade purine and pyrimidine nucleotides, releasing phosphate, carbon, and nitrogen, the latter primarily as ammonium[1,2]. The ammonium is re-assimilated into amino acids[3–5] while most of the carbon is released as ribose, when nucleosides resulting from nucleotide dephosphorylation[6] are hydrolyzed at the N-glycosidic bond[7,8]. For pyrimidine nucleosides, this reaction is catalyzed by nucleoside hydrolase 1 (NSH1), while for purine nucleosides, a complex of NSH1 and nucleoside hydrolase 2 (NSH2) is required[9,10]. NSH1 and NSH2 are cytosolic enzymes[9,11,12] but ribose cannot be further metabolized in the cytosol. To enter the sugar phosphate pool, ribose must be transported into the plastids because ribokinase (RBSK), which phosphorylates ribose to ribose-5-phosphate, is exclusively located there[13,14]. Consequently, a ribose transporter in the inner plastid-envelope membrane is required. The efficient uptake of this sugar into isolated

[1]Leibniz University Hannover, Molecular Nutrition and Biochemistry of Plants, Hannover, Germany. [2]University of Kaiserslautern-Landau, Plant Physiology, Kaiserslautern, Germany. [3]Leibniz University Hannover, Institute of Microbiology, Hannover, Germany. [4]Leibniz University Hannover, Institute of Organic Chemistry, Hannover, Germany. [5]Leibniz University Hannover, Institute of Botany, Hannover, Germany. ✉e-mail: cpwitte@pflern.uni-hannover.de

chloroplasts from *Pisum sativum* (pea) and *Spinacia oleracea* (spinach) was shown decades ago[15,16], but so far the ribose transporter has not been identified on the molecular level.

While all plants can degrade nucleotides, in some tropical legumes like common bean (*Phaseolus vulgaris*) and soybean (*Glycine max*), purine nucleotide catabolism involving ribose release plays a particularly important role when they fix nitrogen in symbiosis with Rhizobia. In root nodules of these plants, the fixed nitrogen is mainly used for the synthesis of purine nucleotides[17], which are then partially degraded to catabolic intermediates, the ureides allantoin and allantoate, that still contain the purine ring nitrogen[18,19]. In these legumes, the ureides serve as long-distance transport compounds for the export of nitrogen from the nodules to the shoot[20,21]. In the shoot, the ureides are fully degraded, releasing ammonium, which is then re-assimilated into amino acids[3,22]. For the biosynthesis of nucleotides, which occurs in the plastids of the nodule, activated ribose is needed[23]. Later, during ureide biosynthesis from the nucleotides, the sugar is released again by the cytosolic NSH1/NSH2 complex[19]. Since ureide biosynthesis is strongly activated in bean nodules, great amounts of ribose are likely released in the cytosol and reused for nucleotide biosynthesis. Thus, an import of ribose into plastids is required to reach RBSK, and this transport is likely to be highly active in this biological context.

So far, only four mono- and disaccharide carriers transporting sucrose, maltose, and glucose have been identified on the molecular level in the inner plastid envelope membrane[24]. Sucrose export from plastids is facilitated by the plastid sucrose transporter (pSuT) belonging to the monosaccharide transporter (MST) like family[25]. The maltose exporter 1 (MEX1) is required for the nocturnal export of maltose, which is the major product of transitory starch breakdown in chloroplasts at night[26]. The loss of MEX1 in *Arabidopsis thaliana* causes an accumulation of maltose and starch, and the plants have a dwarf phenotype[26]. MEX1 complements the *Escherichia coli malF* mutant, deficient in maltose uptake, but a detailed biochemical analysis of MEX1 is still pending[26,27]. The export of glucose from plastids appears to involve the putative plastid glucose transporter (pGlcT)[28], which, as pSuT, belongs to the MST family[29]. Glucose is only a minor product of starch degradation[30], which might explain why Arabidopsis *pGlcT* null mutant plants are phenotypically normal and do not accumulate starch but contain slightly more glucose at night[31]. Interestingly, introducing a *pGlcT* null mutant into the *mex1* background exacerbates the dwarf phenotype of *mex1* plants, while supplying external sucrose attenuates the growth defect of the double mutant. Based on these results, it was suggested that the strong growth phenotype of the *pglct mex1* plants might be caused by severe carbon starvation and that pGlcT might be involved in glucose export from starch breakdown[31]. However, direct evidence for the glucose transport function of pGlcT has not yet been obtained, as the transporter has not yet been characterized biochemically.

Recently, the plastid transporter pGlcT2, which is closely related to pGlcT, has been characterized at the molecular level in Arabidopsis[32]. The analysis of the transport properties in a bacterial system revealed that pGlcT2 is a glucose-specific facilitator with an apparent $K_M$ value of about 3 mM. Ribose, fructose, sucrose, and maltose do not inhibit the glucose transport by pGlcT2, indicating that it is not transporting these sugars[32]. It was suggested that pGlcT2 acts as a glucose importer in young plants to facilitate chloroplast development and that it serves as a glucose exporter upon starch breakdown, especially when plants are grown in extended light phases.

To identify the plastid ribose transporter, we took advantage of the biological context of high purine nucleotide biosynthesis and turnover in *P. vulgaris* nodules. The putative plastid glucose transporter pGlcT of bean (Pv-pGlcT) turned out to be a strong candidate for the ribose transporter. *E. coli* mutants, which were defective in

ribose, glucose, or fructose transport, and expressed *pGlcT* of Arabidopsis (*At-pGlcT*), were used to identify sugars that are transported by pGlcT. Sugar metabolite analyses using T-DNA mutants of Arabidopsis and CRISPR-induced mutants in bean nodules were performed to investigate the in vivo function of pGlcT. Additionally, a possible functional overlap of *At*-pGlcT and *At*-pGlcT2 was assessed by monitoring the growth of Arabidopsis double and triple mutants lacking different combinations of *pGlcT*, *pGlcT2*, and *MEX1*.

## Results

### Identification of a plastid ribose transporter candidate

Purine nucleotide metabolism is particularly active in nitrogen-fixing nodules of common bean (*Phaseolus vulgaris*) and other ureide-exporting legumes like soybean (*Glycine max*)[23,33] but not in roots of these plants or in other legumes that do not produce ureides as nodule-to-shoot nitrogen transport metabolites, for example, *Medicago truncatula* or *Lotus japonicus*. We took advantage of this special biological context following the hypothesis that the missing plastid ribose transporter might be transcriptionally activated in nodules compared to roots of common bean and soybean, but not in nodules of Medicago and Lotus. To identify sugar transporters matching this expression pattern, we analyzed a comparative root and nodule transcriptome dataset of the four legume species that we had recently compiled from publicly available transcriptome data[19]. First, transcripts that were at least 1.5-fold more abundant in common bean nodules compared to roots, based on gene expression data from Kamfwa et al.[34], were compiled, and then nodule and root expression data of corresponding orthologs in the other three legume species were analyzed[19]. Five genes encoding sugar transporters were stronger expressed in nodules than in roots of bean and soybean, but only two of them,' sugar will eventually be exported transporter 7' (*SWEET7*) and the 'plastid glucose transporter' (*pGlcT*), were exclusively induced in nodules of the two bean species (Table 1). Ribokinase (RBSK), required for phosphorylating ribose in the plastids, was also slightly induced in nodules of common bean and soybean but not in the other legumes. By contrast, the expression pattern of *pGlcT2* did not match our search criteria since there was no induction in nodules versus roots in any of the legume species.

SWEET7 lacks a plastid transit peptide and is predicted to reside in the plasma membrane[35]. By contrast, pGlcT contains such a peptide, and the [35]S-labeled protein of Arabidopsis was shown to be imported into isolated spinach chloroplasts[28], which made pGlcT the only transporter that matched all our search criteria. We confirmed the location in plastids by confocal microscopy using mesophyll protoplasts from leaves of *Nicotiana benthamiana* that transiently

**Table 1 | Ratio of transcript abundances in nodules versus roots in legumes for genes of sugar transporters and ribokinase**

| locus | gene | ratio of transcript abundance in nodules versus roots | | | |
|---|---|---|---|---|---|
| | | *P. vulgaris* | *G. max* | *L. japonicus* | *M. truncatula* |
| 002G028300.1 | *PLT6*[a] | 340.1 | 65.1 | 788.8 | 284.4 |
| 004G061900.1 | *pGlcT*[b] | 4.8 | 1.9 | - | - |
| 007G055100.1 | *STP13*[c] | 45.6 | 32.2 | 37.5 | - |
| 009G134300.1 | *SWEET1*[d] | 52.9 | 3.5 | 40.0 | 329.1 |
| 002G300900.1 | *SWEET7* | 141.6 | 2.3 | - | - |
| 005G070000.1 | *RBSK*[e] | 3.3 | 1.5 | - | - |

[a]polyol transporter 6, [b]a second paralog of *pGlcT* encoded at Phvul008G007500 is not induced in nodules compared to roots, [c]sugar transporter protein 13, [d]sugar will eventually be exported transporter 1, [e]ribokinase.
List of *Phaseolus vulgaris* genes and their orthologs of soybean (*Glycine max*), Lotus (*Lotus japonicus*) and Medicago (*Medicago truncatula*). Ratios of transcript abundances are only shown if values are 1.5 or above.

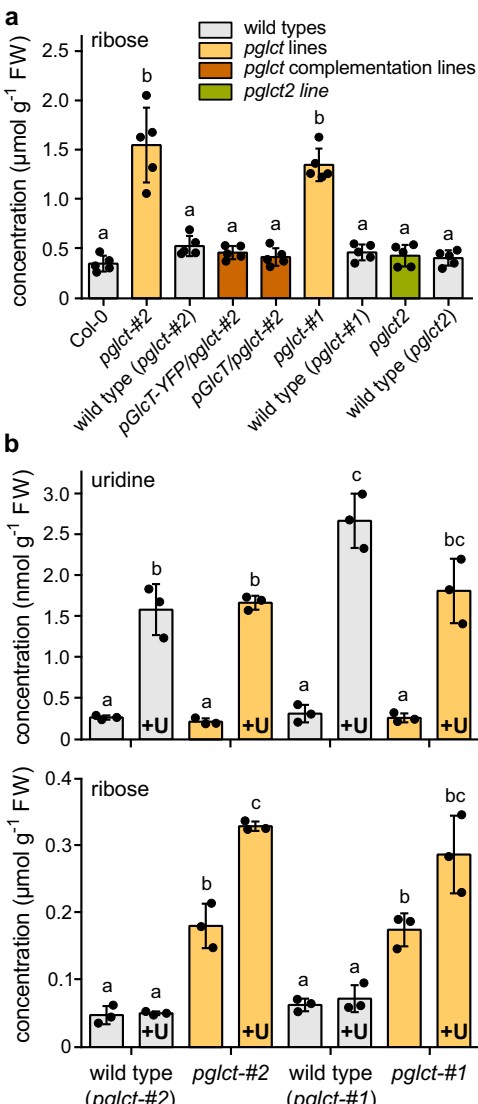

**Fig. 1 | Ribose content in rosettes of Arabidopsis *pGlcT* and *pGlcT2* variants and in liquid culture-grown seedlings treated with uridine. a** Ribose content in rosettes of 28-day-old plants cultivated on soil under long-day conditions (16 h light) after exposure to three days of darkness. Two independent mutants of *pGlcT* (*pglct-#1* and *pglct-#2*) as well as a mutant of *pGlcT2* (*pglct2*) were analyzed. For each mutant, the corresponding wild types were isolated from the respective segregating mutant populations as indicated. An independent wild type (Col-0) and complementation lines expressing either a *pGlcT* or a *pGlcT-YFP* transgene in *pglct-#2* background were also included. Bars show the mean, error bars are SD, n = 5 biological replicates. FW, fresh weight. Statistical analysis was performed with the two-sided Tukey's multiple pairwise comparison test using the sandwich variance estimator[6]. Different letters indicate significant differences at p < 0.05. All p values are listed in the Source Data file. **b** Uridine and ribose contents in seedlings grown in a liquid shaking culture under continuous light for seven days followed by 24 hours of cultivation in new growing media containing 410 μM uridine (labeled +U in the graph) or control media without uridine. Plants were thoroughly washed with distilled water before preparing the material for MS analysis. Bars show the mean, error bars are SD, n = 3 biological replicates. Statistical analysis as in a. Source data are provided in the Source Data file.

expressed Arabidopsis pGlcT (*At*-pGlcT) with a C-terminal yellow fluorescent protein tag (YFP, Supplementary Fig. 1). There is some evidence that *At*-pGlcT is involved in glucose export from plastids during starch breakdown, but it has also been noticed that *pGlcT* is expressed in Arabidopsis and other plant species in non-photosynthetic tissues that in some cases do not accumulate

starch[31,36]. The latter observations suggest a broader function of pGlcT and led us to consider the possibility that pGlcT might be the ribose transporter we were looking for. However, this seemed a daring hypothesis because Schäfer et al.[15] had shown that radioactive glucose uptake into spinach chloroplasts is only weakly inhibited by a fourfold excess of ribose, which suggested that glucose and ribose transport into chloroplasts are not mediated by the same carrier.

### Arabidopsis pGlcT mediates cytosol to plastid ribose transport in vivo

Since pGlcT is conserved in vascular plants[28,32], we used the comparatively easy genetic access to Arabidopsis to test if a mutation in *At*-pGlcT affects the ribose content of leaves in this model plant. Two T-DNA null mutants of *At*-pGlcT (*pglct-#1*, SALK051876; *pglct-#2*, SALK078684) have been described[31]. From segregating populations of these lines, we isolated the homozygous mutants and the corresponding wild types. Complementation lines in the *pglct-#2* background were generated with constructs for the expression of *At*-pGlcT-YFP or untagged *At*-pGlcT, respectively. The lines were screened for the expression of the transgenes by either confocal microscopy or RT-PCR, and two suitable lines were selected (Supplementary Fig. 2). Plants varying in the expression of *At*-pGlcT were grown for 28 days, and then placed in darkness for three days to enhance nucleotide degradation[6,10]. The ribose concentrations in the rosettes of these plants were quantified by liquid chromatography coupled to triple quadrupole mass spectrometry (LC-MS). We also included a mutant of the recently described plastid glucose transporter *At*-pGlcT2[32] and its corresponding wild type in this analysis.

The ribose content of both *At*-pGlcT mutants (*pglct-#1* and *pglct-#2*) was up to 4.4-fold higher than in the other lines, and this molecular phenotype could be complemented by both the *At*-pGlcT and the *At*-pGlcT-YFP transgenes (Fig. 1a). These results demonstrate that At-pGlcT is required for ribose homeostasis and suggest that ribose transport into the plastids to reach RBSK is restricted in the *pglct* background. By contrast, ribose homeostasis was not disturbed in a null mutant of *At*-pGlcT2. (Fig. 1a). This is consistent with the observation that ribose does not affect glucose transport by *At*-pGlcT2[32]. Hence, *At*-pGlcT2 seems to be unable to transport this pentose, which is supported by our in vivo data.

To examine more directly whether a compromised cytosol to plastid ribose transport is the cause for the ribose accumulation in the *pglct* background, we performed a uridine feeding experiment. Uridine is taken up into Arabidopsis cells by the equilibrative nucleoside transporter 3 (ENT3)[37] and hydrolyzed in the cytosol by NSH1 to uracil and ribose[9,10,12]. If ribose import into plastids is compromised in *pglct* plants, we expected the mutants to accumulate more ribose than the wild type in this experiment, because the ribose released in the cytosol from uridine would not be efficiently metabolized by phosphorylation via plastid RBSK in the *pglct* background. Seedling shaking cultures of both *pglct* lines and the corresponding wild types were grown for seven days, then the growth media were exchanged with media containing 410 μM (0.1 mg mL⁻¹) uridine, while the controls contained no uridine. After 24 h, the seedlings were washed thoroughly, and their ribose and uridine contents were determined. The uridine concentration was elevated by treatment with uridine in all genotypes (Fig. 1b upper panel). In the wild types, the uridine treatment did not influence the ribose content (Fig. 1b lower panel), indicating that ribose from cytosolic uridine hydrolysis can be rapidly metabolized. By contrast, the two *At*-pGlcT mutants contained up to 6.5-fold more ribose than the wild types after uridine treatment (Fig. 1b lower panel). Both mutants also had more ribose in the absence of external uridine, and the addition of uridine increased the ribose content further. Also note that the concentration of ribose is about three orders of magnitude higher than that of uridine. These results support the idea

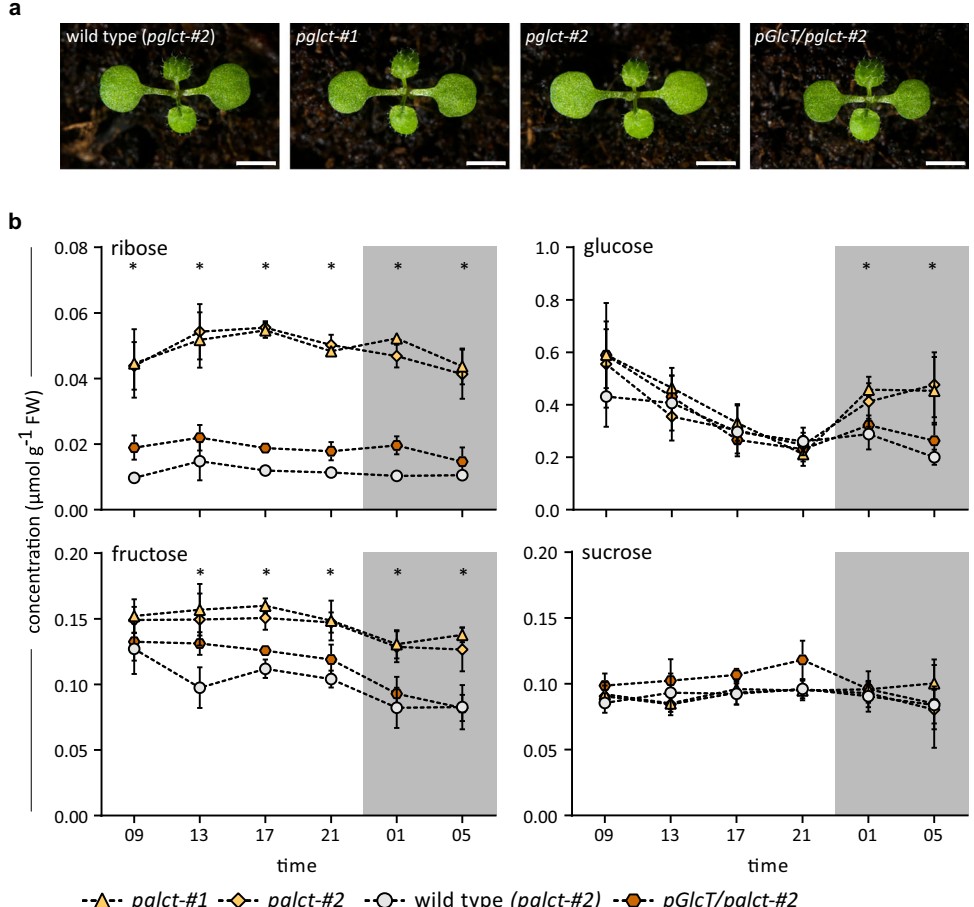

**Fig. 2 | Growth phenotypes and diurnal changes in leaf sugar content of different *At-pGlcT* variants. a** Representative images of the wild type (obtained from segregating *pglct-#2* population), the two mutants (*pglct-#1, pglct-#2*), and the complementation line (*pGlcT/pglct-#2*) in the four-leaf stage after 10 days of growth under long day conditions (16 h light). Scale bar, 2.5 cm. The experiment was repeated three times. **b** Quantification of ribose, glucose, fructose and sucrose in leaves of 10-day-old seedlings (as shown in a). Night hours are indicated by grey shading. Mean values with SD are shown, $n = 5$ for all, except for *pGlcT/pglct-#2* glucose at 1:00 AM ($n = 4$) and ribose, glucose and sucrose at 5:00 AM ($n = 4$) and fructose at 5:00 AM ($n = 3$). Each biological replicate ($n$) was a pool of seedlings grown on soil in an individual pot. Statistical analysis was performed with the two-sided Tukey's multiple pairwise comparison test using the sandwich variance estimator[6]. Asterisks indicate $p$ values < 0.05 between *pglct-#2* and the wild type. Source data and $p$-values are provided in the Source Data file. FW, fresh weight.

that pGlcT of Arabidopsis is required for the efficient transport of ribose from the cytosol into plastids in vivo.

## The homeostasis of ribose, fructose, and glucose is disturbed in *pglct* seedlings

In Arabidopsis *pglct* seedlings grown in shaking culture, ribose accumulated even without an induction of nucleotide degradation by darkness (Fig. 1b), which indicated that ribose homeostasis might always be disturbed in this genetic background. To analyze the *pGlcT* mutants in more detail, several carbohydrates were quantified by gas chromatography coupled to mass spectrometry (GC-MS) in a 24 h time course using 10-day-old seedlings of *pglct-#1, pglct-#2*, the complementation line expressing untagged *pGlcT* (*pGlcT/pglct-#2*), and the wild type. The plants were grown under long-day conditions (16 h light) on soil. The different genotypes all exhibited a wild type phenotype, as previously reported for the *pGlcT* mutants[31] (Fig. 2a). In plants from all genotypes, the ribose concentration remained relatively constant over time but was about twice as high in the mutants than in the complementation line and on average five-fold higher than in the wild type (Fig. 2b). Glucose concentrations continuously decreased during the day and did not differ significantly between the genotypes. With the onset of the night, glucose levels increased, in particular in pGlcT seedlings, similar to what has been reported by Cho et al.[31]. This result is consistent with the release of glucose from starch mobilization in

chloroplasts at night[38,39] and points to a function of pGlcT in glucose export from the plastids. The fructose content of the seedlings decreased slightly at night in all genotypes, but surprisingly, the fructose concentration was significantly elevated in both *pglct* lines at almost all time points. Sucrose contents were nearly constant over time and did not differ between the genotypes. Additionally, nucleotides and nucleosides were quantified to evaluate the potential impact of a disruption of cytosolic ribose removal on nucleotide metabolism. However, neither the content of nucleotides nor nucleosides differed between the distinct genotypes (Supplementary Fig. 3), indicating that cytosolic ribose accumulation does not influence nucleotide metabolism.

## Growth of ribose-uptake deficient *E. coli* with ribose as sole carbon source is supported by pGlcT

To obtain direct evidence for the ribose transport function of pGlcT, we aimed to complement a mutant of *E. coli* defective in ribose transport. *E. coli* can use ribose as the sole carbon source and imports this sugar using the high-affinity ABC-type transporter RbsC[40]. With lower affinity, *E. coli* can also take up ribose employing the allose carrier AlsC[41] and the xylose transporter XylH[42]. Additionally, ribose can be taken up by the glucose phosphotransferase system PtsG, but only if PtsG acquires defined amino acid exchanges[43]. We generated a $\Delta rbsC$ $\Delta xylH$ $\Delta alsC$ triple mutant and tested whether its growth defect

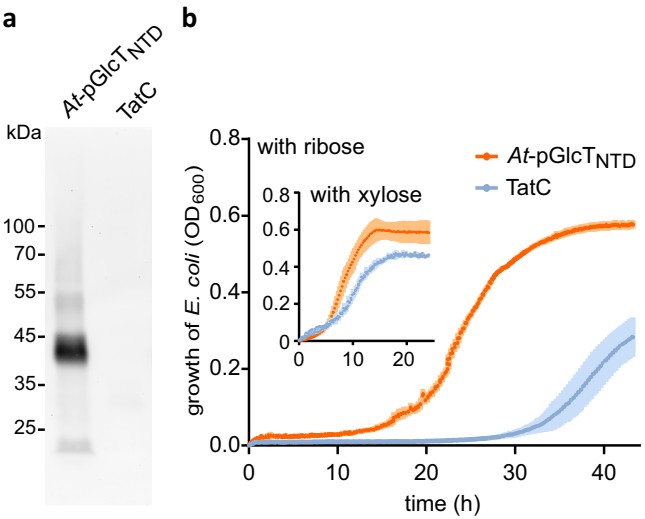

**Fig. 3 | Arabidopsis pGlcT enters the membrane and enhances the growth of an *E.coli* strain compromised in ribose uptake when grown with ribose as the sole carbon source. a** Membrane preparations of *E. coli* MG1655 Δ*ptsG* Δ*manXYZ* expressing an *At-pGlcT* variant (*At*-pGlcT$_{NTD}$) or TatC as a negative control were analyzed by an immunoblot developed with anti-His antibodies. In *At*-pGlcT$_{NTD}$, the N-terminal plastid transit peptide (residues 1 to 82) of *At*-pGlcT is substituted by the N-terminal domain (NTD) of *E. coli* PhoR (residues 1 to 54) to facilitate bacterial membrane targeting. The recombinant protein also has a C-terminal 6xHis tag (for construct details, see the Source Data file). The immunoblot demonstrates that *At*-pGlcT$_{NTD}$ is targeted to the membrane. **b** Growth of *E. coli* BW25113 Δ*rbsC* Δ*xylH* Δ*alsC* expressing *At-pGlcT$_{NTD}$* (orange curves) or the gene of the membrane protein TatC from *E.coli* as a negative control (blue curves). Main figure, growth with 0.4% ribose; inset, growth with 0.4% xylose on M9 minimal media without other carbon sources. The optical density of the culture at 600 nm (OD$_{600}$) was measured. Mean values with SD are shown, *n* = 3 biological replicates. The experiment was performed twice with similar results. Source data are provided in the Source Data file.

on minimal media containing ribose as the sole carbon source can be complemented by *pGlcT* from Arabidopsis.

To this end, an *At-pGlcT* transgene was expressed from the *E. coli* *tatA* promoter for moderate constitutive expression[44]. The produced *At*-pGlcT variant had a C-terminal His tag and lacked the plastid transit peptide (residues 1 to 82), as this is irrelevant for the bacterial system and can interfere with membrane incorporation in *E. coli*[32]. Instead, to foster correct membrane targeting, we added an N-terminal domain (NTD) for membrane-targeting in *E. coli* consisting of the N-terminal region of the *E. coli* membrane protein PhoR[45] (residues 1 to 54). The recombinant protein, which was named *At*-pGlcT$_{NTD}$, entered the membrane of E. coli (Fig. 3a), whereas full-length *At*-pGlcT lacking NTD could not be detected in the membrane (Supplementary Fig. 4). The negative control strain expressed *tatC* encoding a membrane protein of *E. coli* unrelated to sugar transport[46].

With minimal medium containing ribose as the sole carbon source, the control strain required about 20 hours longer to enter the phase of vigorous growth compared to the bacteria that produced At-pGlcT$_{NTD}$, whereas with xylose, both strains grew similarly (Fig. 3b). The result suggests that *At*-pGlcT facilitates ribose import into *E. coli* cells.

**Ribose, glucose, and fructose are transported by pGlcT**
To quantify ribose transport by pGlcT, the time-dependent import of $^{14}$C-ribose into the *E. coli* Δ*rbsC* Δ*xylH* Δ*alsC* triple mutant, either expressing *At-pGlcT$_{NTD}$* or *tatC* as a control, was analyzed (Fig. 4a). Bacteria expressing *At-pGlcT$_{NTD}$* imported ribose at a higher rate than the negative control, confirming that pGlcT transports ribose. Still, the control strain was able to import ribose, albeit at a substantially lower

rate, probably due to side activities of other (sugar) transporters. This result is consistent with the ability of the control strain to grow slowly in minimal media containing ribose (Fig. 3b).

Next, the dependence of the ribose uptake on the substrate concentration was measured, calculated as the difference in uptake to the negative control (Fig. 4b). An apparent K$_M$ value for ribose of $1.22 \pm 0.27$ mM was determined (V$_{max}$ of $20.86 \pm 1.78$ nmol $10^9$ cells$^{-1}$ h$^{-1}$). Additionally, the ribose uptake was measured at different external pH values showing that the uptake activity of the strain expressing *At-pGlcT$_{NTD}$* was significantly higher than that of the control strain at all pH values (Fig. 4c). The activity was strongest between pH 6 to 8 with a peak at pH 7, which corresponds to typical pH values occurring in the plant cytosol and the chloroplast stroma[47]. The results indicate that the ribose transport activity of *At*-pGlcT is independent of a pH gradient. To test the substrate specificity of *At*-pGlcT, the uptake of radioactive ribose (1 mM) was measured in the presence of a 10-fold excess of different non-radioactive mono- and disaccharides as potential competitors (Fig. 4d). As expected, excess ribose suppresses the uptake of radioactively labeled ribose to the background level of the negative control, but also glucose was an equally efficient suppressor supporting the postulated role of pGlcT as plastid glucose exporter. Surprisingly, fructose might be another substrate of pGlcT as its presence also reduced the uptake of the radioactively labeled ribose (Fig. 4d). Maltose and mannose also appeared to reduce ribose uptake slightly, but there is a higher probability that this might have been observed by chance ($p > 0.05$).

To determine if pGlcT transports glucose and fructose, we measured the uptake of radiolabeled glucose or fructose in other monosaccharide uptake mutants of *E. coli* expressing *At-pGlcT$_{NTD}$* or *tatC* as a control. We either used an *E. coli* strain deficient in glucose and mannose uptake (MG1655 Δ*ptsG* Δ*manXYZ*)[48] or a strain lacking the high-affinity fructose transport system FruA (BW25113 Δ*fruA*)[49]. In time course uptake experiments using bacteria of the respective genetic background and 1 mM labeled glucose or fructose, the bacteria that expressed *At-pGlcT$_{NTD}$* imported significantly more glucose or fructose compared to the *tatC* controls (Fig. 4e and f). Unfortunately, it was not possible to determine the dependence of glucose and fructose transport on the respective substrate concentration because the background uptake rates were generally too high, in particular at higher substrate concentrations. Nonetheless, the results show that pGlcT can transport glucose and fructose. The observed fructose and glucose accumulation in *pglct* seedlings of Arabidopsis (Fig. 2) is therefore likely a direct consequence of the *pGlcT* defect, resulting in disturbed monosaccharide exchange over the inner plastid envelope membrane.

**Stunted growth of *mex1* Arabidopsis plants is aggravated by additional mutation of *pGlcT* but not *pGlcT2***
Both pGlcT and pGlcT2[32] are glucose transporters of the inner plastid envelope. Are they mutually redundant in supporting glucose export from starch breakdown? If they were, the reported strong growth reduction of *pglct mex1* plants resulting from sugar starvation[31] might also be observed in *pglct2 mex1* plants, and this phenotypic peculiarity should be aggravated in a triple mutant lacking all three transporter genes. We tested this hypothesis with plants grown under long-day conditions (16 h light) on soil and documented the phenotypes at 12 and 34 days after sowing (Fig. 5, Supplementary Fig. 5). All seedlings germinated evenly and initially grew similar to the wild type. When the first true leaves had developed after 12 days, it became apparent that *mex1* plants were smaller than the wild type but this was more severe in *pglct mex1* and *pglct pglct2 mex1* seedlings (Fig. 5a). The differences were quantified by measuring the leaf area and the ratio of yellow to green leaf pixels in the leaf images (Fig. 5b). The results show that the *pglct2 mex1* plants were not different from the *mex1* plants and that the triple mutant had a similar phenotype to the *pglct mex1* line. Also, later in development, after 34 days, no additional phenotypic differences

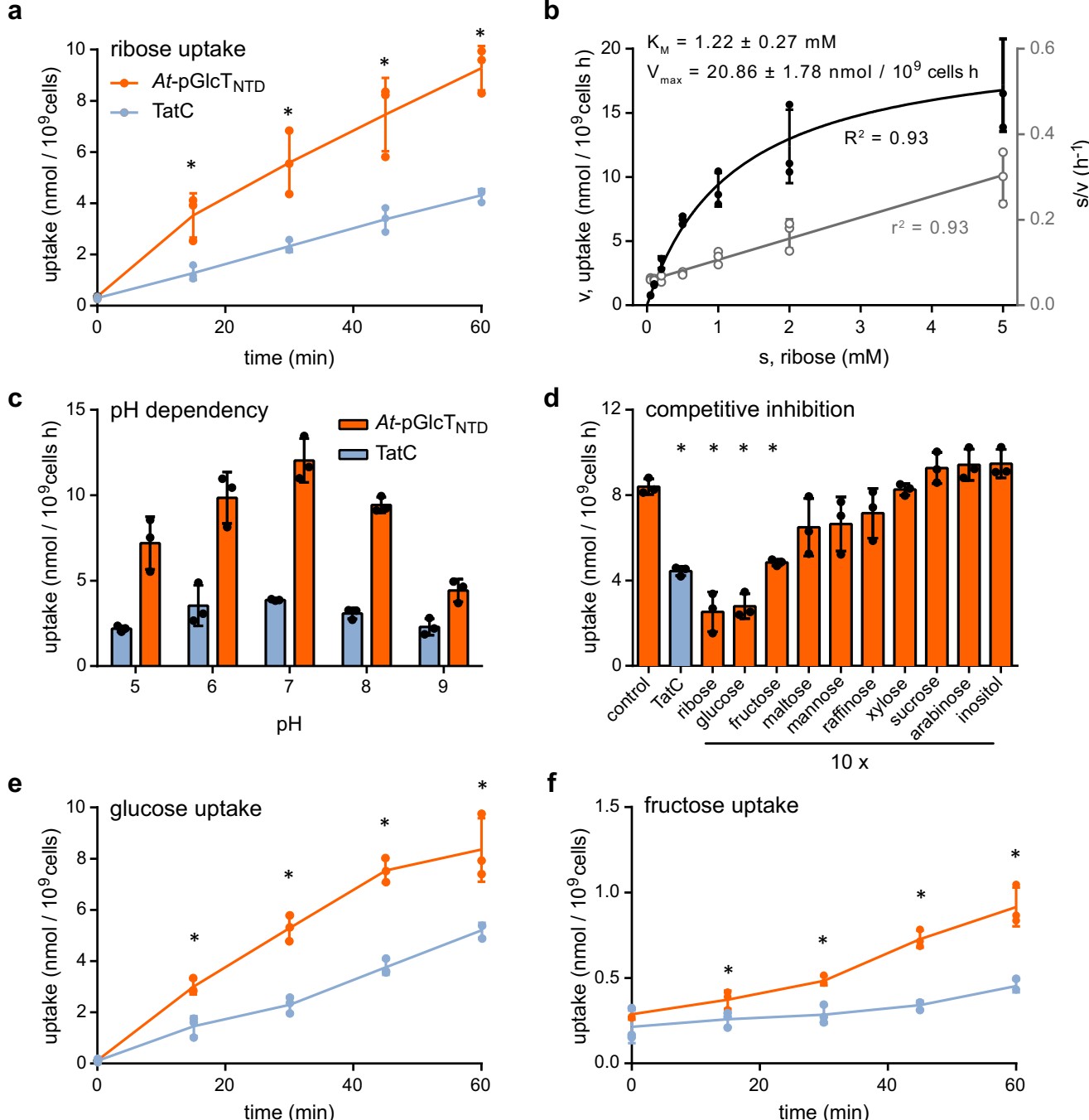

**Fig. 4 | Transport activity of *At*-pGlcT in *E. coli* mutants defective in ribose, glucose or fructose transport. a** Time-dependent ribose uptake in *E. coli* BW25113 Δ*rbsC* Δ*xylH* Δ*alsC* producing *At*-pGlcT$_{NTD}$ (orange) and the membrane protein TatC from *E. coli* as a negative control (blue curves). *E. coli* cells were incubated with 1 mM of ribose containing 0.1 µCi of $^{14}$C-ribose at pH 7 for the indicated time intervals. **b** Dependency of ribose import on the substrate concentration in *E. coli* expressing *At*-pGlcT$_{NTD}$. Uptake was determined for different concentrations of $^{14}$C-ribose at pH 7 and calculated as the difference between import in cells expressing *At*-pGlcT$_{NTD}$ and *tatC* after 30 minutes of incubation. The dependency of velocity (v) on the substrate concentration (s) was plotted and fitted with the Michaelis-Menten equation (black, left y-axis). The data were linearized according to Hanes and fitted by linear regression (grey, right y-axis). **c** The dependency on pH of ribose import into *E. coli*. Cells expressed either *At*-pGlcT$_{NTD}$ (orange bars) or *tatC* (blue bars) and were incubated with 1 mM labeled ribose (as above) and incubated

for 30 minutes. **d** Substrate specificity of *At*-pGlcT$_{NTD}$. Binding of different sugars to *At*-pGlcT$_{NTD}$ was determined by competitive inhibition of $^{14}$C-ribose uptake (1 mM initial outside concentration) in the presence of non-radioactive sugars in ten-fold excess at pH 7 after 30 minutes of incubation. **e** Time-dependent glucose uptake activity was measured in *E. coli* MG1655 Δ*ptsG* Δ*manXYZ* producing *At*-pGlcT$_{NTD}$ (orange) or TatC (blue). *E. coli* cells were incubated with 1 mM of glucose containing 0.2 µCi of $^{14}$C-glucose at pH 7 for the indicated time intervals. **f** Time-dependent fructose uptake activity was measured in *E. coli* BW25113 Δ*fruA* producing *At*-pGlcT$_{NTD}$ (orange) or TatC (blue). *E. coli* cells were incubated with 1 mM of fructose containing 1 µCi of $^{3}$H-fructose at pH 7 for the indicated time intervals. Mean values with SD are shown, *n* = 3 biological replicates. Asterisks indicate *p* values < 0.05 between cells expressing *At*-pGlcT$_{NTD}$ and *tatC* (A-C, E and F) and between the positive control and any other test condition (D) calculated using the two-sided Student´s t-test. Source data and *p*-values are provided in the Source Data file.

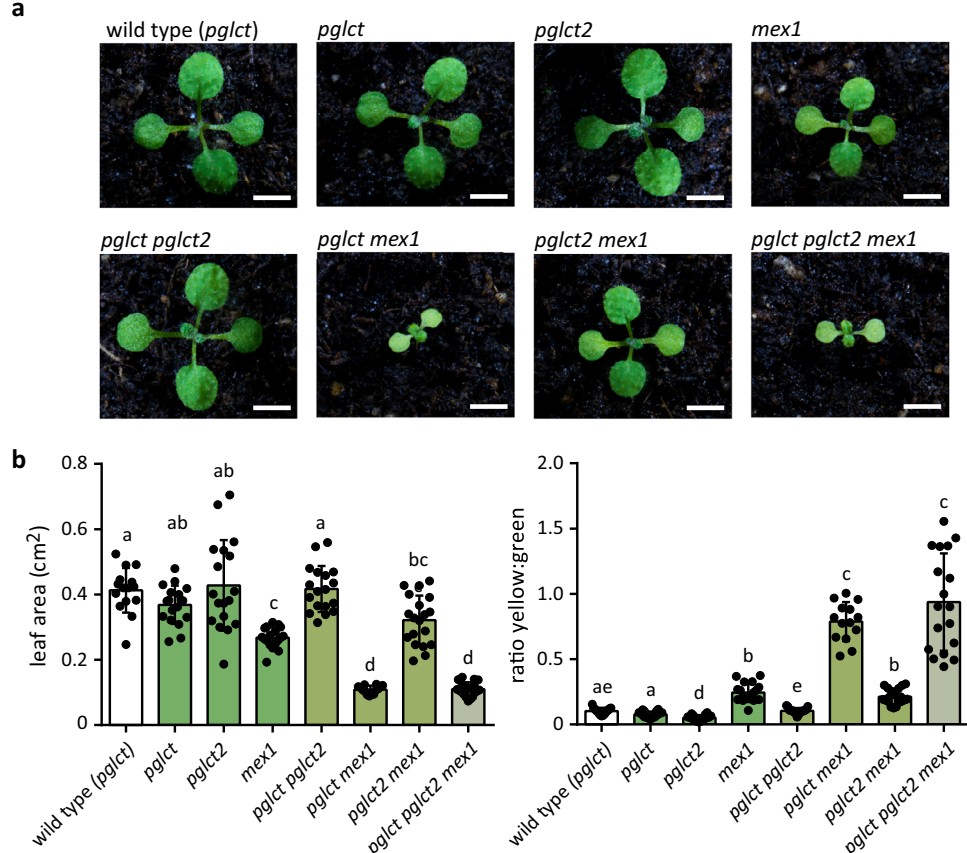

**Fig. 5 | Phenotypic analysis of Arabidopsis plants with single or combined defects in the two plastid glucose carriers (pGlcT, pGlcT2) and in the plastid maltose transporter (MEX1). a** Representative images of 12-day-old seedlings of the indicated genotypes grown under long-day conditions (16 h light). Crosses involving *pGlcT* were performed with plants containing the *pglct-#2* allele. The wild type was obtained from a segregating *pglct-#2* population. Scale bar, 1 cm. **b** Quantification of the leaf area (left panel) and the proportion of yellow to green pixels in the leaf images of 12-day-old seedlings as shown in a. A replicate (*n*) is one seedling grown on soil in a pot. Mean values and SD are shown with *n* = 15 for WT, *n* = 18 for *pglct*, *pglct2*, *mex1* and *pglct pglct2*, *n* = 14 for *pglct mex1*, *n* = 19 for *pglct2 mex1* and for *pglct pglct2 mex1*. The statistical analysis was performed with the two-sided Tukey's multiple pairwise comparison test using the sandwich variance estimator[6]. Different letters indicate significant differences with *p* value < 0.05. Source data and *p* values are provided in the Source Data file.

were observed (Supplementary Fig. 5). We conclude that pGlcT is probably the main glucose exporter downstream of plastid starch breakdown and that there is no obvious redundancy between pGlcT and pGlcT2, at least under the chosen growing conditions.

**In bean nodules, *Pv*-pGlcT plays a role in ribose and fructose homeostasis**

The hypothesis that pGlcT might transport ribose had originally been derived from the expression analysis of sugar transporter genes in legume nodules (Table 1). To investigate whether ribose recycling is important for ureide biosynthesis in common bean nodules, transgenic nodules mutated either in *pGlcT* or in *RBSK* of *P. vulgaris* were generated. There are two *pGlcT* paralogs in *P. vulgaris*. One is induced in nodules compared to roots (Phvul.004G061900, Table 1), reaching an RPKM (reads per kilobase per million mapped reads) of 91 in nodules in the RNA-seq dataset we analyzed[3,4] whereas the other (Phvul008G007500) is not induced and is expressed with an RPKM of only 16 in nodules. We aimed to mutate the induced gene that we called *Pv-pGlcT;a* to distinguish it from the non-induced gene (*Pv-pGlcT;b*). For this, hairy roots were generated with *Agrobacterium rhizogenes* carrying CRISPR/Cas9 constructs for gRNA-directed mutagenesis. Later, nodule formation was induced on the transgenic roots by inoculation with *Rhizobium tropici*. Nodules that had developed on hairy roots expressing an empty vector without encoded gRNAs served as control. Carbohydrates were analyzed by GC-MS[50], and LC-MS was used for enhanced nucleotide and nucleoside analysis[51,52]. The

recovery of metabolites was generally over 70% (Supplementary Table 1), which validated this LC-MS method for the analysis of nodule tissue.

Out of ten independent transgenic roots analyzed for each mutant, four were null mutants in *Pv-pGlcT;a* and three in *Pv-RBSK* (Supplementary Tables 2 and 3, Supplementary Fig. 6). Nodules from these roots were used for metabolite analysis.

As in Arabidopsis seedlings, ribose and fructose concentrations were higher in *pglct;a* nodules than in the control (Fig. 6a). However, glucose concentrations were similar, maybe because starch was not degraded when the nodules were harvested at about mid-day. Interestingly, in *rbsk* nodules, more ribose accumulated than in *pglct;a* nodules. The mutation of *RBSK* leads to a complete block of ribose recycling[14], which does not seem to be the case in *pglct;a* nodules indicating that *Pv*-pGlcT;b or other transporters can also facilitate the import of ribose into plastids and can partially compensate for the lack of *Pv*-pGlcT;a. Also in Arabidopsis, we observed a higher ribose accumulation in *rbsk* background compared to *pglct* (Supplementary Fig. 7). Sucrose and its hydrolysis product fructose were slightly more abundant in *pglct;a* nodules. This might be due to lower sucrose consumption in *pglct;a* background caused by reduced ureide output which, however, was not reflected in lower steady-state concentrations of ureide biosynthesis intermediates and the end product allantoin (Fig. 6a, b). In *rbsk* nodules, the steady-state contents of some ureide biosynthesis intermediates were lower than in the control (IMP and uric acid), and the amount of allantoin was strongly reduced. Fully

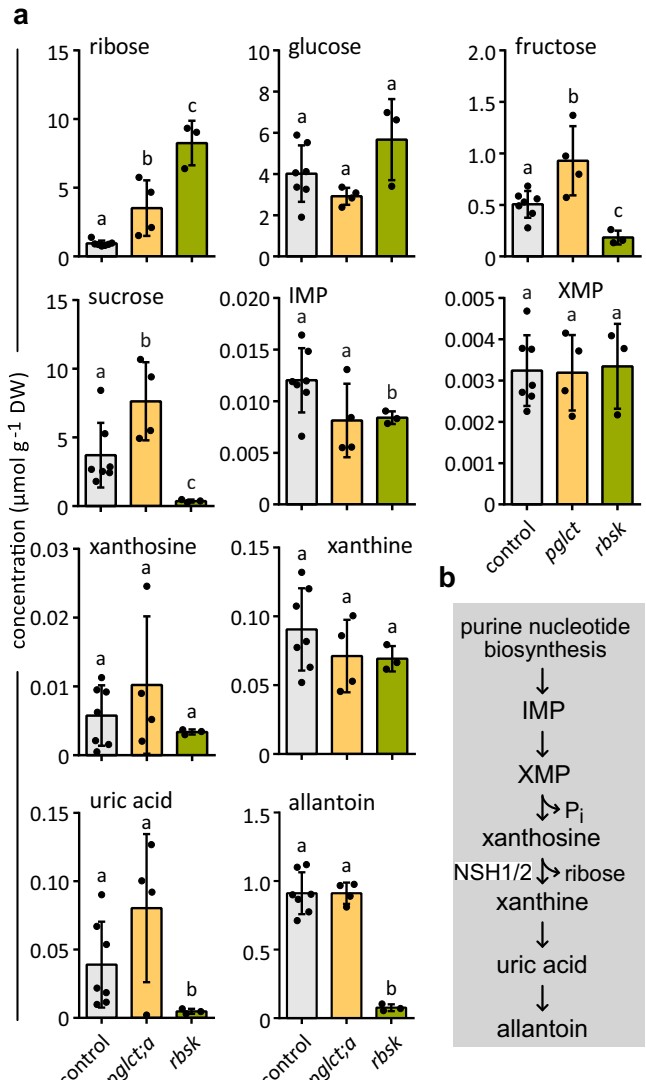

**Fig. 6 | The content of sugars and ureide biosynthesis metabolites in mutant bean nodules lacking *Pv*-pGlcT or *Pv*-RBSK. a** The content of ribose, glucose, fructose, and sucrose as well as intermediates of ureide biosynthesis and the end product allantoin in null-mutant bean nodules defective in *pGlcT* ($n = 4$) or *RBSK* ($n = 3$) versus empty-vector controls ($n = 7$). A biological replicate ($n$) is a nodule pool from a single transgenic root with a unique mutation event (see Supplementary Tables 2 and 3 and Supplementary Fig. 6). Additionally, all nodule samples came from independent plants that were grown together. Mean values with SD are shown. Concentrations are given in reference to nodule dry weight (DW), which is about 20% of the fresh weight. The statistical analysis was made with the two-sided Tukey's multiple pairwise comparison test using the sandwich variance estimator[6]. Different letters indicate $p$ values < 0.05. Source data and $p$ values are provided in the Source Data file. **b** Overview scheme showing the metabolites of the ureide/allantoin biosynthetic pathway. NSH1/2, the complex of nucleoside hydrolase 1 (NSH1) and nucleoside hydrolase 2 (NSH2).

cause product inhibition of nucleoside hydrolysis by NSH1 that is an essential part of the NSH1/NSH2 complex required for xanthosine hydrolysis in the ureide biosynthesis pathway[10,19] (Fig. 6b). However, xanthosine does not accumulate in the mutants (Fig. 6a) maybe because its production is already reduced when ribose recycling for purine nucleotide biosynthesis is disturbed.

If *Pv*-pGlcT;a is primarily involved in ribose recycling for purine biosynthesis in bean nodules, the carrier should be well expressed in infected nodule cells, because purine nucleotide biosynthesis occurs mainly in this cell type[18,53]. We fused the potential promoter and the 5'-UTR upstream of the *Pv*-pGlcT;a translation start codon (altogether a 3 kb region) to the coding sequence of a peroxisome-targeted mNeonGreen-SKL as a fluorescent reporter in a binary vector for expression in hairy roots and nodules. In nodules expressing this promoter-reporter construct, mNeonGreen fluorescence was primarily observed in infected cells and far less in uninfected cells of the infection zone (Fig. 7, Supplementary Fig. 9). Assuming that the site of promoter activity reflects the location of the corresponding protein, this result is in agreement with a primary role of *Pv*-pGlcT;a in ribose recycling for purine nucleotide biosynthesis in infected cells of bean nodules.

## Discussion

Here, we demonstrate that pGlcT functions as a ribose carrier, mediating the import of ribose from cytosolic nucleoside hydrolysis into plastids (Figs. 1, 2, 4, 6). By contrast, pGlcT2 cannot transport ribose[32], which fits with our observation that there is no ribose accumulation in *pglct2* Arabidopsis plants (Fig. 1a). In the plastid, the ribose is phosphorylated by RBSK to ribose-5-phosphate[13,14]. This reaction is the only known metabolic sink for ribose in plants. Ribose-5-phosphate can either directly enter the Calvin cycle in photosynthetically active chloroplasts or the oxidative pentose phosphate pathway (OPPP) active in plastids in the dark[54]. Alternatively, ribose-5-phosphate can be further phosphorylated by PRPP synthetase (PRS) to 5-phosphoribosyl-1-pyrophosphate (PRPP) (Fig. 8a). PRPP is required in plastids for purine nucleotide biosynthesis, which is particularly active in plastids of the infected cells in bean and soybean nodules[18,53,55,56]. Alternatively, PRPP can be consumed for the synthesis of histidine[57] or tryptophan[58] in the plastids (Fig. 8a).

Our data also demonstrate that pGlcT can transport glucose, which supports previous biochemical, genetic and metabolic results[28,31], suggesting that pGlcT contributes to glucose export from plastids during nocturnal starch breakdown (Fig. 8b). Although pGlcT2 is also a plastid glucose carrier[32], pGlcT2 does not appear to be redundant to pGlcT regarding the export of glucose from starch degradation (Fig. 5). In many tissues of Arabidopsis, pGlcT is far more abundant than pGlcT2[59], which further supports the notion that pGlcT is the main glucose carrier in the plastid envelope. However, a possible functional overlap between pGlcT and pGlcT2 for plastid glucose homeostasis should be investigated in greater detail under various growth conditions and including molecular analyses.

Ribose transport by pGlcT is probably not energized by a pH gradient (Fig. 4c). Generally, pGlcT appears to facilitate monosaccharide transport in both directions over the membrane depending on the given concentration gradient of the sugar. Hence, glucose and fructose could also be imported into plastids by pGlcT, for example in non-photosynthetic tissues for starch synthesis or to support the OPPP (Supplementary Fig. 10). Glucose and fructose can both be metabolized in plastids entering the hexose phosphate pool by phosphorylation mediated by the plastid hexokinase (pHXK encoded at At1g47840) and the plastid fructokinase (FRK3 encoded at At1g66430)[60]. *FRK3* is ubiquitously expressed whereas *pHXK* is mainly expressed in the root and during germination[61], indicating that glucose phosphorylation is only required in non-photosynthetic plastids. It has been demonstrated, however, that non-photosynthetic plastids rely

blocking ribose recycling appears to impair allantoin production, which might then also cause a halt of sucrose import into the nodule indicated by the low sucrose and fructose contents of *rbsk* nodules (Fig. 6a).

However, apart from potentially reduced purine nucleotide biosynthesis due to defects in ribose recycling, *pglct;a* and especially *rbsk* nodules might also suffer from inhibition of ureide biosynthesis caused by ribose accumulation. The NSH1 substrates inosine and uridine as well as the uridine precursor cytidine accumulated when the nodule ribose content was high (Supplementary Fig. 8). Ribose might

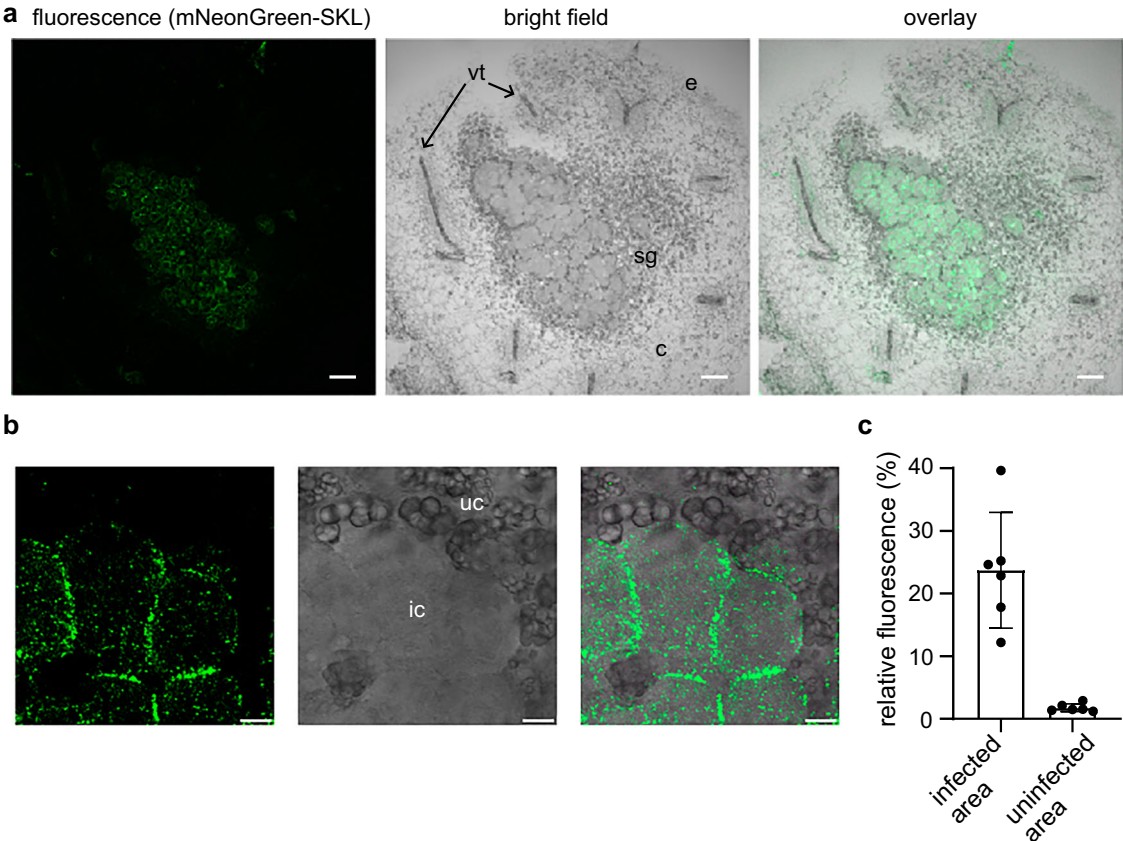

**Fig. 7 | Activity of the *Pv-pGlcT;a* promoter of *P. vulgaris* in nodules. a** Confocal fluorescence microscopy image of a cross-section from a nodule expressing peroxisomal *mNeonGreen-SKL* under the control of a 3 kb *Pv-pGlcT;a* promoter and the 5′-UTR fragment upstream of the bean *Pv-pGlcT;a* gene translation start codon. Left panel, fluorescence channel; middle panel, bright field channel; right panel, overlay of both channels. In the bright field image, areas with infected cells appear gray and smooth, whereas areas with uninfected cells contain dark starch granules (sg) that only occur in this cell type. vt, vascular tissue; e, epidermis; c, cortex. Scale bars, 100 µm. **b** Close-up view of a nodule cross-section in the infection zone showing neighboring infected cells (ic) and uninfected cells (uc) with starch granules. Channels as in a. Scale bars, 10 µm. **c** Quantification of relative fluorescence (ratio of green to black pixels) in infected and uninfected areas of the infection zone using six independent cross sections from two independent nodules. Mean values with SD are shown. The images that were used for quantification are shown in b and in Supplementary Fig. 9. Source data are provided in the Source Data file.

strongly on the import of glucose-6-phosphate[62–65] suggesting that the direct import of glucose or fructose and their phosphorylation in the plastid can at best be a minor contribution to the hexose phosphate pool of non-photosynthetic plastids. In agreement with this, there is no reduction in starch accumulation in the root tip of Arabidopsis mutants lacking the plastid phosphoglucose isomerase 1 (PGI1)[66] that interconverts fructose-6-phosphate and glucose-6-phosphate (Supplementary Fig. 10). This suggests that fructose import into plastids, followed by phosphorylation and conversion to glucose-6-phosphate, is not contributing significantly to starch synthesis in the Arabidopsis root tip. Nonetheless, the fructose transport by pGlcT is probably of physiological relevance since we observed constitutively higher fructose concentrations in *pglct* seedlings of Arabidopsis (Fig. 2b) and slightly more fructose in bean nodules mutated in *pGlcT;a* (Fig. 6).

In bean nodules, the complete inhibition of ribose recycling in *rbsk* nodules strongly reduces the concentration of allantoin, indicating that the flux towards the ureides is compromised (Fig. 6). Such a drastic effect was not observed in nodules with a defect in *pGlcT;a* indicating that some ribose can still be imported into plastids in this genetic background. While *pGlcT;a* is induced and strongly expressed in nodules, the paralog (*pGlcT;b*) is expressed at a lower constitutive level. Nonetheless, it can probably also mediate ribose transport into plastids. However, pGlcT;a seems to be the isoform dedicated to ribose recycling due to its predominant expression in infected cells (Fig. 7, Supplementary Fig. 9), where purine nucleotide biosynthesis mainly

takes place. A similar expression pattern was observed for a *pGlcT;a* ortholog of soybean using single-cell transcriptomics of the nodule[53]. In Arabidopsis, there is only one *pGlcT* gene, but still more ribose accumulates in *rbsk* than in *pglct* seedlings (Supplementary Fig. 7). Hence, other plastid (sugar) transporters can probably also mediate ribose transport with lower affinity.

It is likely that pGlcT also contributes to the export of glucose from starch breakdown in bean nodules, where starch predominantly accumulates in uninfected cells[67,68]. While *pGlcT;a* in common bean and its ortholog in soybean are mainly expressed in infected cells (Fig. 7)[53] for ribose transport, the ortholog of *pGlcT;b* in soybean is expressed in uninfected and cortex cells[53], possibly to support glucose export from starch turnover in plastids. Consistent with this idea, we did not observe glucose accumulation in *pglct;a* nodules (Fig. 6a).

We have shown here that pGlcT facilitates ribose and glucose transport over the inner plastid envelope membrane. Interestingly, the two sugars are transported in opposite directions in the respective physiological context (Fig. 8a, b). We have shown that fructose is also a substrate of pGlcT. Furthermore, the accumulation of fructose in Arabidopsis leaves and bean nodules in mutants of *pGlcT* indicates that fructose is transported by pGlcT in vivo. While the metabolic purpose of ribose and glucose transport is clear, the physiological significance of the fructose exchange between the cytosol and the plastids remains to be investigated.

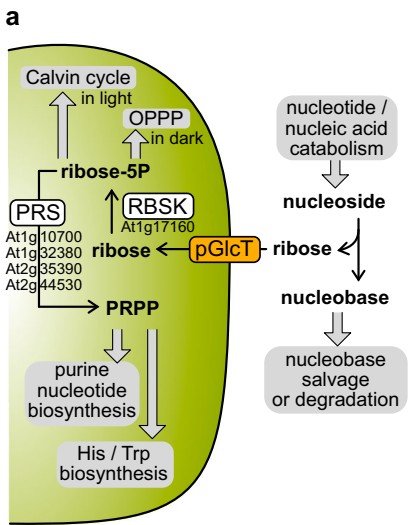

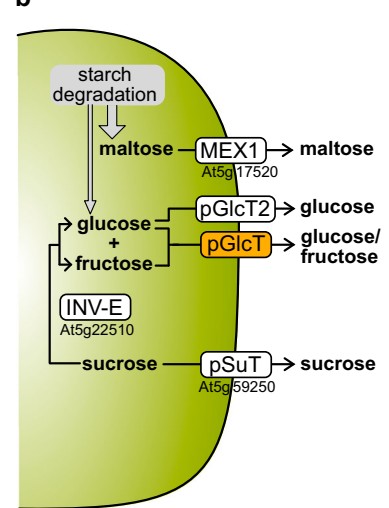

**Fig. 8 | Models for the distinct physiological roles of pGlcT. a** Function of pGlcT in ribose import into plastids to reach ribokinase (RBSK) which catalyzes the only ribose-consuming reaction in plant metabolism. After phosphorylation, ribose-5-phosphate can directly enter the Calvin cycle (chloroplasts), the oxidative pentose phosphate pathway (OPPP, in plastids in the dark), or be converted by PRPP synthetase (PRS) to 5-phosphoribosyl-1-pyrophosphate (PRPP), which is required for purine nucleotide biosynthesis and the biosynthesis of histidine and tryptophan. All three biosynthesis pathways reside in plastids. **b** Function of pGlcT in glucose export during starch degradation and possibly in export of glucose and fructose resulting from hydrolysis of sucrose by plastid invertase (INV-E). Glucose is only a minor product of starch degradation which produces mainly maltose exported via MEX1. The main carrier for glucose export appears to be pGlcT and not pGlcT2. Sucrose can alternatively be exported by pSuT instead of being hydrolyzed by INV-E. A third model for a putative role of pGlcT in glucose and fructose import into plastids is shown in Supplementary Fig. 10.

## Methods

### Plant material and experimental conditions

The T-DNA null-mutants of *pGlcT* (At5g16150) in Col-0 background, *pglct-#1* (SALK051876) and *pglct-#2* (SALK078684), that were described by Cho et al.[31], and the T-DNA null-mutant of *pGlcT2* (At1g05030, *pglct2*, SALK052078), that was described by Valifard et al.[32], were obtained as segregating lines from the Nottingham Arabidopsis Stock Centre. Homozygous mutants and corresponding wild types were obtained from the segregating populations using the gene-specific primers P733 and P734 for *pglct-#1*, P735 and P736 for *pglct-#2*, and P1199 and P1200 for *pglct2* together with primer N61 binding on the T-DNA. A list of all primers can be found in the Source Data file. The *rbsk-1* line (SALK007531) described by Schröder et al.[14] was used as *RBSK* mutant. Complementation lines of *pglct-#2* were generated by transformation with pXCS-pGlcT-YFP (construct H391) and pXCScpmv-pGlcT (construct H392). Sequences and maps of the constructs can be found in the Source Data file. The complementation lines were screened by confocal microscopy (for H391) or by RT-PCR (for H392) using leaf RNA, a polyT primer for reverse transcription and primers P1156 and P1157 for amplification of the cDNA derived from the transgene. As a control, the cDNA of *Actin2* (At3g18780) was amplified with primers 1033 and 1034.

Arabidopsis and *N. benthamiana* plants were grown under long-day conditions (16 h light of 85 μmol m$^{-2}$ s$^{-1}$ at 22 °C and 8 h night at 20 °C) with 60% relative humidity. Surface-sterilized seeds were grown on soil (75% peat, 25% perlite, pH 5.5, and full nutrition, Stecklings-medium, Klasmann-Deilmann, Geeste, Germany). For the Arabidopsis seedling liquid culture, 50 sterilized seeds were grown under sterile conditions in a 50 mL Erlenmeyer beaker containing 10 mL of growth medium (2 mM $KNO_3$, 1 mM $NH_4NO_3$, 1 mM glutamine, 1 mM $MgSO_4$, 60 μM $H_3BO_3$, 14 μM $MnSO_4$, 1 μM $ZnSO_4$, 0.6 μM $CuSO_4$, 0.3 μM $Na_2MoO_4$, 40 μM sodium iron EDTA, 4 mM $CaCl_2$, 3 mM $KH_2PO_4$ / $K_2HPO_4$, 3 mM MES, 0.5% sucrose, adjusted to pH 5.8). The culture was stratified at 4 °C overnight and then grown for seven days at 22 °C with shaking (50 rpm for three days followed by 80 rpm for four days under constant light) in an Innova 42 shaker (New Brunswick) equipped with a light bank. To supply uridine, the culture medium was exchanged with the same medium but containing 0.1 mg mL$^{-1}$ (410 μM) uridine and the growth continued for 24 h. In controls, a medium without uridine was used. For growing *P. vulgaris* (cultivar Negro Jamapa), seeds were surface sterilized and cultivated in a mixture of vermiculate and perlite (1:2) under long day conditions (16 h light of 125 μmol m$^{-2}$ s$^{-1}$ at 28 °C and 8 h night at 25 °C) with 65% relative humidity. Growth, transformation of the seedlings, selection of transgenic roots, inoculation with *Rhizobium tropici* CIAT899 and harvest of the transgenic nodules were performed exactly as described by Voß et al.[19].

### Cloning

The coding sequence of Arabidopsis *pGlcT* without a stop codon was amplified from leaf cDNA with the primers P1012 and P1013, introducing flanking *ClaI* and *XmaI* sites for cloning. The same fragment, including a stop codon, was amplified with the primers P1012 and P1014. The PCR products were cloned into pJET1.2 (Thermo, K1231), resulting in pJET-pGlcT1 (H388) and pJET-pGlcT2 (H389). The inserts from the constructs H388 and H389 were released with *ClaI* and *XmaI* and cloned into pXCS-eYFP (V36)[69] and pXCScpmv-HAStrep (V69)[70], respectively, generating pXCS-pGlcT-YFP (H391) and pXCScpmv-pGlcT (H392). Maps and sequences of constructs H391 and H392 can be found in the Source Data file.

As the plastid transit peptide had to be removed for the bacterial system, we initially amplified the gene without the N-terminal transit peptide coding region (residues 1 to 80), using the primers pGlcT-NdeI-F and pGlcT-ClaI-R, and cloned the fragment into pABS-*pspA*-H6[71], using the *NdeI* and *ClaI* restriction sites. Two further N-terminal codons were deleted via QuikChange (Stratagene), using the primer glcT-del-F and its complementary reverse primer, finally resulting in pABS-pGlcT-H6. To achieve efficient insertion of pGlcT into the bacterial cytoplasmic membrane, the region encoding the N-terminal domain (NTD) of *E. coli* PhoR, comprising the first two transmembrane helices (amino acids 1 to 54), was cloned upstream of the *GlcT* coding sequence, resulting in pGlcT$_{NTD}$. For this, the respective *phoR* region was PCR-amplified with genomic *E. coli* DNA as template using the

primer pair phoR-NdeI-F and phoR-AseI-R. The PCR product was then restricted with *NdeI* and *AseI* and cloned into the *NdeI*-linearized vector pABS-pGlcT-H6, resulting in pABS-pGlcT$_{NTD}$-H6 (H1865, Source Data file). The control construct pABS-tatC (H1579) was described before[32]. All constructs were confirmed by sequencing.

A 2828 bp genomic fragment upstream of the Phaseolus *pGlcT;a* (locus Phvul.004g061900) start codon containing the 5′-UTR and presumably the promoter was amplified with the primers P2835 and P2836 introducing flanking *AscI* and *XhoI* sites, cloned into pJET1.2 (pJET-pGlcTprom-Pv, H1416), excised from H1416 with these enzymes and cloned into pY2CS-mNeonGreenSKL (V197)[19] replacing the 35S promoter with the *pGlcT* promoter (pY2C-pGlcTprom-PV-mNeon-GreenSKL, H1420). In this binary vector for hairy root transformation, the promoter drives the expression of a *mNeonGreen* reporter gene encoding the C-terminal peroxisomal targeting sequence SKL.

To induce CRISPR/Cas9-mediated mutations in Phaseolus root nodules, constructs were generated by GoldenGate-cloning using the MoClo-system[72,73] precisely as described by Voß et al.[19]. Fragments encoding individual parts of the gRNA together with the scaffold were first amplified with the primers P293 and P272, in combination with a guide-specific forward and reverse primer from pGTR as described[74]. Guide-specific forward and reverse primers to generate CRISPR constructs targeting Phaseolus *pGlcT* (Phvul.004G061900) and *RBSK* (Phvul.005G070000) were P1729 and P1730 and P1733, and P1734, respectively. All following steps were performed as described in Voß et al.[19], resulting in CRISPR constructs encoding Cas9 directed against *pGlcT;a* (H1187) and *RBSK* (H1190). The CRISPR constructs also encode a green fluorescent protein (GFP) reporter for the selection of transgenic hairy roots and nodules. Transformation of bean seedlings with *A. rhizogenes* carrying the construct H1078 without a guide array[19] served to obtain negative control nodules without mutations.

## Bacterial mutant strains

The Keio-collection strains BW25113 *rbsC::kan*, BW25113 *xylH::kan* and BW25113 *alsC::kan*[49] were used to construct the triple deletion strain BW25113 Δ*rbsC* Δ*xylH* Δ*alsC* by two successive phage transductions using P1vir according to standard protocols. After the initial selection on LB agar plates containing kanamycin (50 µg mL$^{-1}$), recipient clones were purified, and the position of the kanamycin cassette was confirmed by colony PCR. To permit P1 transduction with strains already generated by previous P1 transductions, the kanamycin cassette was removed using pCP20-endcoded flippase according to the standard protocol of Datsenko & Wanner (2000)[75]. The loss of the kanamycin cassette was confirmed via colony PCR. For the indicated experiments, also a Δ*ptsG* Δ*manXYZ* strain was used[48], or BW25113 *fruA::kan*[49].

## *E. coli* growth analyses

The triple mutant of *E. coli* BW25113 Δ*rbsC* Δ*xylH* Δ*alsC* has strongly reduced ribose uptake capability. This strain was transformed with pABS-pGlcT$_{NTD}$-H6 (H1865) for the expression of different variants of pGlcT. The same strain expressing *tatC* from pABS-tatC (H1579)[32] served as a negative control. Each strain was grown in three biological replicates. For each replicate, an LB pre-culture containing 25 µg mL$^{-1}$ chloramphenicol (Cm) and inoculated with a single colony was grown at 37 °C until an OD$_{600}$ of 1.0 was reached. M9 minimal medium (5 mL, Supplementary Table 4) containing Cm and 0.4% (w/v) xylose as the sole carbon source was inoculated with 20 µL of pre-culture and grown overnight at 37 °C (180 rpm). The bacteria were pelleted at 3000 g for 3 min and washed twice with 0.5 mL M9 minimal medium including Cm and 0.4% (w/v) ribose as the sole carbon source before diluting them to a final optical density of 0.02 in this medium. For monitoring growth in a SpectraMax iD3 Microplate reader (Molecular Devices), bacteria were grown at 37 °C under constant shaking (517 rpm) in sterile 96-well plates (Sarstedt, 821.581.001) in a final volume of 200 µL

per well. Lids were fixed with superglue to the plates to avoid abrasion of the plastic.

For detection of pGlcT in the cell membrane of *E. coli*, strain MG1655 Δ*ptsG* Δ*manXYZ* expressing *pGlcT$_{NTD}$* or *tatC* was cultivated aerobically at 37 °C in LB medium (1% (w/v) tryptone, 1% (w/v) NaCl, 0.5% (w/v) yeast extract), supplemented with 25 µg mL$^{-1}$ chloramphenicol (Cm). For protein detection, 100 mL LB medium was inoculated to an OD$_{600}$ of 0.1 with an overnight culture. Cell densities were normalized to an OD$_{600}$ of 1 after an incubation of 3.5 h, and cells were harvested by centrifugation (3260 g, 4 °C, 10 min). Cell pellets were resuspended in 20 mM Tris HCl (pH 8.0) and after adding DNase I and 1 mM phenylmethylsulfonylfluorid (PMSF), cells were homogenized using a French press (two passages, 800 p.s.i.). Cell debris was removed by low-speed centrifugation (16,060 g, 4 °C, 20 min) and membranes were separated from the soluble fraction by ultracentrifugation (130,000 g, 4 °C, 30 min). Membrane pellets were resuspended in 20 mM Tris HCl pH 8.0, and analyzed by SDS–PAGE and immunoblot using standard methods. For detection, mouse monoclonal anti His-tag antibodies (A00186, lot No 19C001747, 1:2,500; GenScript, Netherlands) were used in combination with a goat polyclonal anti-mouse IgG secondary antibody coupled to horseradish peroxidase (31430, lot No YC371618, 1:10,000; Pierce Biotechnology, USA) for enhanced chemiluminescence (ECL) detection. Images of immunoblots were acquired utilizing the Intas Advanced Imager system (Intas, Germany).

## Uptake assays

Liquid cultures of *E. coli* Δ*rbsC* Δ*xylH* Δ*alsC* expressing *pGlcT$_{NTD}$* or *tatC* were grown overnight at 37 °C and 180 rpm in LB media supplemented with Cm. These cultures were used to inoculate fresh cultures to an OD$_{600}$ of 0.2, which were then incubated until an OD$_{600}$ of 1 was reached. Cells were pelleted (5 min, 3000 g) and washed with uptake buffer (15 mM MES, 15 mM HEPES, 15 mM Tris, 5 mM MgCl$_2$, pH 7) before resuspension to an OD$_{600}$ of 5 in uptake buffer. Time-dependent uptake studies were started by the addition 1 mM ribose labeled with 0.1 µCi $^{14}$C-ribose (Hartmann-Analytic). The reaction was incubated at 37 °C and 200 rpm for a total of 60 min. Aliquots of 100 µL bacterial suspension were taken at 0, 15, 30, 45, and 60 min and the bacteria were collected on a 0.45 µm MCE membrane filter (Merck Millipore, HAWG01300). The filters were washed by vacuum filtration with uptake buffer to remove excess ribose and transferred to scintillation vials containing 4 mL Rotiszint eco plus (Carl Roth, 0016.3) prior to scintillation counting (Tri-Carb 4810 TR, PerkinElmer).

For concentration-dependent uptake assays and the determination of K$_M$, import assays were performed as stated above, but the uptake reaction was initiated by the addition of ribose at final concentrations of 0.05, 0.1. 0.2, 0.5, 1, 2, 5 mM, respectively. The pH dependency was assessed in uptake buffer adjusted to the indicated pH values by addition of HCl or NaOH and using a ribose concentration of 1 mM in the reaction mix. The transport specificity was tested by the addition of a 10x excess concentration (10 mM each) of the putative competitive carbohydrates over the labeled ribose (1 mM). For the analysis of the concentration- and pH-dependency as well as the transport specificity, the reactions were stopped after 30 minutes by collecting the *E. coli* cells via vacuum filtration.

For the uptake kinetics with radioactive glucose and fructose, *E. coli* defective in the endogenous glucose and mannose (MG1655 Δ*ptsG* Δ*manXYZ*) or fructose transport systems (BW25113 Δ*fruA*) were used. These strains expressed either *pGlcT$_{NTD}$* or *tatC* as a control. Time-dependent uptake was analyzed as described above using either 1 mM glucose containing 0.2 µCi $^{14}$C-glucose (Hartmann-Analytic) or 1 mM fructose containing 1 µCi $^{3}$H-fructose (Hartmann-Analytic) as substrates.

## Protoplast and nodule preparation for confocal microscopy

To obtain protoplasts from *N. benthamiana* expressing *pGlcT-YFP* from Arabidopsis (H391), leaf strips of 1 mm were vacuum infiltrated for 30 min in the dark with an enzyme solution consisting of 1.5% (w/v) cellulase (Yakult Pharmaceutical, L0012), 0.4% (w/v) macerocyme (Yakult Pharmaceutical, L0021), 400 mM mannitol, 20 mM KCl, 10 mM CaCl$_2$ and 0.1% bovine serum albumin in 20 mM MES buffer. The infiltrated leaf strips were incubated for 3 h in darkness.

Protoplasts were analyzed using a Leica TSC SP8 microscope equipped with an HC PL APO CS2 ×40/1.10 WATER objective (Leica Microsystems). YFP was excited at 514 nm with 0.15% laser power, and the emission was collected between 520 and 537 nm using 10% gain of a HyD detector. Auto fluorescence of the chloroplasts was measured between 658 and 708 nm using a gain of 584 with a photomultiplier detector.

Transgenic nodules expressing pY2C-pGlcTprom-PV-mNeonGreenSKL (H1420) were cut with a Leica vibratome VT1000S into 60 μm slices. Images of nodule cross sections were taken with an HC PL FLUOTAR 10 × 0.30 dry (whole nodule) or HC PL APO CS2 × 40/1.10 WATER objective (magnification of the central infected region, Leica Microsystems) and processed using the Leica Application Suite X (Leica Microsystems). The excitation of GFP and mNeonGreen was done at 488 nm with 0.5% laser power, and the emission was recorded from 502 to 537 nm. Control wild-type nodules were analyzed in parallel with the exact same settings to ensure that fluorescence signals were not derived from background.

## Amplified fragment length polymorphism

Genomic DNA was extracted from nodules, and AFLP analyses were performed as described in Voß et al.[19]. Primer pairs for amplification of fragments including the predicted gRNA binding region were P1787 and P1788 for *pGlcT;a*, and P1791 and P1792 for *RBSK*.

## Phenotype analysis

The leaf area and the ratio of yellow and green pixels were determined as described[76]. Images were taken with a Nikon DS-Ri2 camera installed on a Nikon SMZ25 stereo microscope. The chlorotic leaf area and green leaf areas were quantified using Fiji[77] with ImageJ 1.53c using Hue Color Threshold settings of 34 to 58 for chlorotic areas and 59 to 90 for green areas. The sum of all pixels was used to calculate the leaf area.

## Metabolite analyses

The quantification of ribose and uridine by LC-MS in the Arabidopsis seedlings grown in the liquid shaking culture (Fig. 1 and Supplementary Fig. 7) was performed exactly as described by Schröder et al.[14], extracting 100 mg of fresh plant material per repeat.

Potentially edited hairy root nodules were identified by GFP fluorescence. All nodules from a single transgenic root were collected in a 2 mL safe seal reaction tube. Freeze-dried nodules were homogenized with one 10 mm steel bead using a mixer mill MM 400 (Retsch) for 1 min at 18 s$^{-1}$. Frozen shoots from 10 day-old Arabidopsis seedlings were homogenized in a 2 mL safe seal reaction tube including four 5 mm steel beads using the same mixer mill for 2 min at 28 s$^{-1}$.

The method for enhanced nucleotide and nucleoside analyses was conducted exactly as described recently[51,52]. For nodule tissue, 1 mg freeze-dried and homogenized material was used. 0.2 μL of DL-allantoin-5-$^{13}$C,1-$^{15}$N isotope standard (1 mg mL$^{-1}$; CDN Isotopes, L340P13) was included during the extraction. One half of the nucleoside-containing fraction was used for nucleoside quantification as reported before, while the second half was freeze-dried and the pellet dissolved in 1.5x the original volume of acetonitrile: 50 mM NH$_4$Ac (pH 5.8, ratio 95:5) for quantification of allantoin and uric acid (Supplementary Table 5). For chromatography, the SeQuant ZIC-cHILIC 3 μm, 100 Å 150 × 2.1 mm (Merck Millipore, 150658) was used. Solvent A was acetonitrile: 50 mM NH$_4$Ac (pH 5.8, ratio 95:5). Solvent B was dH$_2$O, 50 mM

NH$_4$Ac (pH 5.8) and acetonitrile (ratio 50:45:5). The injection volume was 10 μL and the flow rate 0.3 mL min$^{-1}$. The solvent B gradient was 0.0 min, 0%; 5.0 min, 5%; 10.0 min, 25%; 15.0 min, 30%, 17.0 min, 65%; 20.0 min, 95%; 31.0 min, 95%; 31.1 min, 5%; 40.1 min, 5%.

Note that allantoate cannot be quantified this way as it is degraded during the metabolite extraction process. Additionally, UMP cannot be quantified because partial hydrolysis of the abundant UDP-glucose during extraction increases the comparatively small UMP pool[78].

To calculate the isotope recovery from nodule tissue, six 2 mL safe seal reaction tubes each containing 1 mg of freeze-dried and homogenized nodule material, were prepared. Three samples were processed as described before, and in three samples, the isotope standards were omitted during the extraction but added later, prior to vacuum centrifugation or freeze drying. Mean peak areas of isotope-labeled metabolites from both types of samples were used to calculate the recovery rate for each metabolite from nodule tissue (Supplementary Table 1).

Samples for carbohydrate analyses by gas chromatography coupled to mass spectrometry (GC-MS) were prepared as described by Lisec et al.[50] with some adaptations. Specifically, metabolites were extracted from either 50 mg of homogenized frozen shoots of Arabidopsis seedlings or 5 mg of homogenized freeze-dried common bean nodules with 750 μL 100% methanol ($-20$ °C), followed by vigorous vortexing. Isotope standard mix (3 μL) was added per sample, which contained 5-$^{13}$C-ribose (Cambridge Isotope Laboratories, CLM-1066), U-$^{13}$C$_6$-fructose (CLM-1553), U-$^{13}$C$_6$;1,2,3,4,5,6,6-D$_7$-glucose (CDLM-3813), and sucrose (glucose-α-1,4-$^{13}$C$_6$-fructose; CLM-9811), each at 0.1 mg mL$^{-1}$. After vortexing, samples were incubated at 70 °C and 950 rpm for 10 min before centrifugation at 11000 *g* for 10 min. 700 μL of the supernatant was transferred to a 2 mL safe seal reaction tube, and 375 μL chloroform ($-20$ °C) and 750 μL water (4 °C) were added. Samples were vortexed, followed by centrifugation at 2200 *g* for 10 min. 300 μL of the upper polar phase was transferred into a new 1.5 mL safe seal reaction vial and evaporated using a vacuum centrifuge. For derivatization, 40 μL of methoxyamine hydrochloride (20 mg mL$^{-1}$, Merck, 89803) in pure pyridine (TCI, Q0034) was added per sample. Samples were incubated at 37 °C and 950 rpm for 2 h. Next, 70 μL MSFTA reagent (Macherey & Nagel, 701270.110) was added to each sample, which was further incubated at 37 °C and 950 rpm for 30 min. Samples were transferred to glass vials suitable for GC-MS analysis (Macherey & Nagel, 702891 and 702730).

The injection volume was 1 μL at 230 °C with a helium carrier gas flow of 1.25 mL min$^{-1}$. The chromatography column was Optima 5 HT (30 m × 250 μm, particle size 0.25 μm; Macherey & Nagel, 726106.30). The start temperature was 80 °C with an increase of 12 °C min$^{-1}$ to 360 °C. The temperature was maintained for 2 min before cooling down. The metabolites were detected using an Agilent 5977 GC/MSD system. The ion source temperature was set to 230 °C, and the recorded mass range was m/z 30 to m/z 650. The instrument was automatically tuned according to the manufacturer's recommendations. The chromatograms were analyzed using the Agilent Chemstation MSD data analysis tool. The GC-MS parameters for quantification are listed in Supplementary Table 6.

## Statistics and Reproducibility

For the comparison of multiple groups, the statistical analyses were performed with the two-sided Tukey's multiple pairwise comparison test using the sandwich variance estimator[6]. Two-sided t-tests were used in Fig. 4 for the comparison of two groups. Replicate numbers were chosen as high as technically feasible. For soybean mutant experiments, the number of biological replicates was limited by the number of null mutants obtained by the CRISPR/Cas9 mutagenesis of the hairy roots. Metabolite samples were analyzed blinded and randomized. Plants for phenotypic analyses were grown in a randomized setup. No data were excluded from the analyses.

## Reporting summary

Further information on research design is available in the Nature Portfolio Reporting Summary linked to this article.

## Data availability

The datasets generated and analyzed as part of this study are included in this article (and its supplementary information files). Source data are provided with this paper. Accession codes: Arabidopsis: *pGlcT*, At5g16150 [https://www.uniprot.org/uniprotkb/B9DGP1/entry]; *pGlcT2*, At1g05030 [https://www.uniprot.org/uniprotkb/Q0WVE9/entry]; *RBSK*, At1g17160 [https://www.uniprot.org/uniprotkb/A1A6H3/entry]; *MEX1*, At5g17520 [https://www.uniprot.org/uniprotkb/Q9LF50/entry]. *P. vulgaris* (v2.1 annotation): *pGlcT;a*, Phvul.004G061900 [https://phytozome-next.jgi.doe.gov/report/gene/Pvulgaris_v2_1/Phvul.004G061900]; *pGlcT;b*, Phvul.008g007500 [https://phytozome-next.jgi.doe.gov/report/gene/Pvulgaris_v2_1/Phvul.008G007500]; *RBSK*, Phvul.005G070000 [https://phytozome-next.jgi.doe.gov/report/gene/Pvulgaris_v2_1/Phvul.005G070000]. Source data are provided with this paper.

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

## Acknowledgements

We thank Hildegard Thölke for technical assistance. This work was supported by the Deutsche Forschungsgemeinschaft (DFG) grants WI3411/7-1 (no. 452173586) and INST 187/741-1 FUGG (no. 423879281) to C.-P.W. Work in the lab of H.E.N. was financially supported by the DFG within the Collaborative Research Center 175, The Green Hub, Project B03.

## Author contributions

C.-P.W. devised and coordinated the project and supervised the experiments in plants and the *E. coli* growth assays. E.N. and I.K. designed and performed the radioactive uptake assays. T.B. and D.M.-B. designed and generated the *E. coli* ribose uptake triple mutant and the constructs for $pGlcT_{NTD}$ expression and provided expertise for the *E. coli* growth assays and for work with transmembrane proteins in *E. coli*. M.H. performed the comparative transcriptomics. L.V. analyzed the Arabidopsis mutants in a time course assisted by G.D. for the GC-MS measurements, performed the *E. coli* growth assays, analyzed the Arabidopsis crosses, and generated and analyzed the bean nodule mutants. N.M.-E. performed the bean *pGlcT;a* promoter study with constructs from L.V. J.R. and J.F. analyzed the nodule image data. R.S. generated and characterized the Arabidopsis mutants and analyzed ribose and uridine contents in the genetic variants together with A.S. L.V. performed the statistical analyses. L.V. and C.-P.W. wrote the manuscript, which was revised by all authors.

## Funding

## Competing interests

The authors declare no competing interests.
