## [Transparent Peer Review file · Nature Communications]

A plastid carbohydrate carrier mediates ribose recycling from nucleotide catabolism and glucose export from starch degradation

Corresponding Author: Professor Claus-Peter Witte

Version 1:

Reviewer comments:

Reviewer #1

(Remarks to the Author)

This study elucidates the role of Arabidopsis pGlcT and its legume ortholog as plastidial ribose transporters and explores their functional significance in symbiotic nitrogen fixation within root nodules. The work employs a multifaceted approach combining plant mutants, bacterial complementation assays, and CRISPR-edited legume nodules. The findings provide novel insights into the intersection of carbon metabolism and symbiotic nitrogen fixation, which fills the gap in the role understanding of how ribose enters chloroplasts from cytosol. Here, some issues require clarification to improve before publication.

1. The lack of ribose accumulation in Arabidopsis pglct2 mutants suggests functional divergence from pGlcT. To clarify this in legumes: Please provide transcriptomic datasets showing pGlcT2 expression in soybean/common bean roots/nodules.
2. What is the subcellular localization of intact pGlcT when it is expressed in E. coli? How does the author determine that 1-82 amino acids are plastid localization signal peptides? What is the subcellular localization result in E. coli after adding N-terminal domain (NTD) for membrane-targeting in E. coli? Please supplement the relevant data.
3. In the growth and development analysis of Arabidopsis mutants with different combinations of pglct, mex1 and pglct2, the author found that pglct2 did not seem to affect its growth, so does it mean that pglct-specific ribose or fructose transport is more important in Arabidopsis growth than pglct2 ?
4. The title of the paper is rather perplexing, and I would suggest revising it to something more comprehensible.
5. The text involves pGlcT from different species, please include the Latin name abbreviations for each to distinguish them clearly, for example AtpGlcT.
6. Line 31-32, to the best of my knowledge, so far, only the proton-uncoupled members of the SWEET family have been shown to transport sugars bidirectionally. However, can pGlcT, as a proton-coupled sugar transporter, also facilitate bidirectional transport?
7. pGlcT is the name of plastid glucose transporter family from MST superfamily, I suggest rename this gene, like pGlcT1, 2, 3 or pGlcT-like 1.
8. Line 33,99,141, In vivo or in vitro needs italics.
9. Line 145, what is 'their corresponding wild types', I do not understand, please clarify the differences between them and the wild-type Arabidopsis.
10. To facilitate comparison, please standardize the units for ribose content and avoid using multiple notations within the same article, like ng mg⁻¹ FW, μmol g⁻¹ FW, μmol g⁻¹ DW.
11. To distinguish with the pglct2 mutant, it would be preferable to rename the pGlcT mutant to pglct#2.
12. The writing quality of this manuscript needs to be enhanced to ensure it is accessible and understandable to the broad readership of Nature Communications.

Reviewer #2

(Remarks to the Author)

This manuscript aims to address an intriguing question about how ribose, derived from nucleotide degradation, is imported

into plastids for recycling into metabolic processes. Researchers found that the plastid glucose transporter pGlcT is a potential player, although pGlcT transports glucose, fructose, and ribose. They also demonstrated the role of pGlcT in nodule production. However, due to the absence of a strong phenotype in the pglct mutant, the biological significance of pGlcT remains questionable. In other words, other transporters, rather than pGlcT, may contribute significantly.

Major comments here:

- 1) The direct evidence supporting that pGlcT transports ribose, glucose, and fructose was obtained in *E. coli*. We know that proteins may not be folded the same way as in eukaryotic cells. It would be stronger if the authors could provide additional evidence supporting this in eukaryotic cells.
- 2) Since a concentration of glucose 10 times higher can completely suppress ribose transport activity in substrate competition experiments (Fig. 4d), and the glucose content in leaves is approximately 10 times greater than that of ribose (Fig. 2), ribose may never have been transported via pGlcT in vivo.
- 3) Quantitative data are required for Fig. 7.
- 4) It is unusual to depend on only one positively identified transgenic plant and to have such low expression in constitutive expression lines (Supplemental Fig. 2).

Reviewer #3

(Remarks to the Author)

In this work, the authors identified and functionally characterized the first plastid ribose transporter in plants. The work is substantive and highly significant to the sugar transporter field. As the authors described in the manuscript, only a small number of plastid sugar transporters have been identified so far, which has hampered our understanding of the exchange of sugars across the inner plastid membrane. The authors first demonstrated the transport function of pGlcT for ribose, glucose and fructose in a *E. coli* sugar-import mutant and then confirmed its in vivo function of the plastid ribose transporter in both *Arabidopsis* and bean, further strengthening their conclusion. I believe the authors have provided sufficient evidence to demonstrate that pGlcT is a plastid facilitator for the import of ribose from nucleotide catabolism, for the export of glucose from nocturnal starch breakdown, and for cytosol-plastid fructose exchange in vivo.

I only have two minor points for the authors to consider when revising the work.

1. As shown in Fig 4d, both glucose and fructose are competitors for ribose transport. It would be very helpful to readers if the authors could use published concentrations of glucose and fructose (and ribose if available) in the cytosol and plastid to give readers an idea on how effective pGlcT is in transporting ribose vs. glucose and fructose in vivo.
2. The diagram presented in Fig 8 has 3 parts. I feel it could be simplified by removing Fig 8C and incorporate the transport of glucose and fructose from the cytosol to the plastid into Fig 8a and provide a couple sentences in the figure legend to indicate this possibility.

Version 2:

Reviewer comments:

Reviewer #1

(Remarks to the Author)

Reviewer #2

(Remarks to the Author)

Thank the authors for responding to my comments. But I still have some following up questions.

1. While I agree that *E. coli* has been widely used as a heterologous system to characterize transporter activity, this does not eliminate all concerns. A recent study (Andersen, 2025, PNAS, PMID: 40279393) demonstrates that different heterologous systems can yield substantial differences in kinetic parameters—up to a 30-fold difference for the same transporter. Notably, the targeting peptide was removed in this study, which could also impact substrate specificity. Therefore, I believe it is reasonable to question whether the results from *E. coli* accurately reflect the transporter's activity in planta.
2. The authors propose that pGlcT may function bidirectionally, but no direct evidence is provided to support this claim. Does this mean that pGlcT symports ribose and glucose into the chloroplast during the day but imports ribose and exports glucose at night? If so, what is the proposed mechanism that supports such bidirectional transport with different directions of substrates?
3. The lack of a strong growth phenotype generally suggests that the gene's biological significance may be limited under the tested conditions—though this does not necessarily mean the gene is functionless. Still, based on the conditions the authors specifically used, the contribution of pGlcT appears modest.

4. According to Valifard et al. (2023, PMID: PMC10318453), pGlcT2 is not expressed in mature leaves and is only expressed in very young leaves. Given this, do the authors still believe pGlcT2 mediates glucose export from starch degradation in mature leaves? Is the pGlcT2 mutant a valid control for assessing ribose homeostasis in Fig 1? The authors should provide metabolite measurements from tissues where pGlcT2 is normally expressed in the wild type.
5. Please recheck the replicate data for uric acid in pglct in Fig 6a; one replicate appears to have a value even lower than any of the control replicates, which may not make sense. Is this an outlier? If excluded, the average uric acid level in pglct might actually be higher than in the control. How do the authors interpret this result in the context of purine metabolism?
6. The labeling of the complementation lines is unclear. The notation "pglct-#2:pGlcT" suggests that the pGlcT gene is under the control of its own promoter, but it reads as if a promoter is driving a gene in a reporter construct. A clearer format would be pGlcT/pglct-#2 to indicate the complementation genotype.

Reviewer #3

(Remarks to the Author)

The authors have addressed all the concerns I raised in the initial review.

REVIEWER COMMENTS

Reviewer #1 (Remarks to the Author):

This study elucidates the role of Arabidopsis pGlcT and its legume ortholog as plastidial ribose transporters and explores their functional significance in symbiotic nitrogen fixation within root nodules. The work employs a multifaceted approach combining plant mutants, bacterial complementation assays, and CRISPR-edited legume nodules. The findings provide novel insights into the intersection of carbon metabolism and symbiotic nitrogen fixation, which fills the gap in the role understanding of how ribose enters chloroplasts from cytosol. Here, some issues require clarification to improve before publication.

1. The lack of ribose accumulation in Arabidopsis *pglct2* mutants suggests functional divergence from pGlcT. To clarify this in legumes: Please provide transcriptomic datasets showing pGlcT2 expression in soybean/common bean roots/nodules.

REPLY: The comparative transcriptomic data in Table 1 shows all sugar transporter genes that matched our search criteria (stronger expression in nodules than in roots). pGlcT2 is not shown in that table because it is not stronger expressed in nodules compared to roots in the data we used. We have added a sentence to the first paragraph of the results section to clarify this: "By contrast, the expression pattern of *pGlcT2* did not match our search criteria since there was no induction in nodules versus roots in any of the legume species." In the publication that recently described pGlcT2 from Arabidopsis (Valifard et al., 2023 PMID: 37088133), the glucose transport could not be inhibited by a 10-fold excess of ribose. This suggests, that pGlcT2 cannot transport ribose and is indeed functionally different from pGlcT.

2. What is the subcellular localization of intact pGlcT when it is expressed in *E. coli*? How does the author determine that 1-82 amino acids are plastid localization signal peptides? What is the subcellular localization result in *E. coli* after adding N-terminal domain (NTD) for membrane-targeting in *E. coli*? Please supplement the relevant data.

REPLY: We have now added new data in Fig. 3a which demonstrate that pGlcT_{NTD} is present in the plasmamembrane of *E. coli*. Intact full-length pGlcT cannot be detected in the membrane or in inclusion bodies of *E. coli*. We suppose the protein is unstable and rapidly degraded. We now show the corresponding data in the new Supplementary Fig. 4.

The TargetP web server (<https://services.healthtech.dtu.dk/services/TargetP-2.0/>) predicts a plastid localization peptide on pGlcT with a high score (73% likelihood) and predicts a cleavage site for this peptide between the amino acids 82 and 83. In a multiple alignment of pGlcT sequences from various plants, this position (82 in pGlcT from Arabidopsis) corresponds approximately to the end of an N-terminal region with low primary sequence conservation, which is typically observed for plastid localization peptides. Therefore, the amino acids 1 to 82 were removed.

3. In the growth and development analysis of Arabidopsis mutants with different combinations of *pglct*, *mex1* and *pglct2*, the author found that *pglct2* did not seem to affect its growth, so does it mean that *pglct*-specific ribose or fructose transport is more important in Arabidopsis growth than *pglct2* ?

REPLY: It has been shown that pGlcT2 only transports glucose but probably not ribose or fructose (Valifard et al., 2023 PMID: 37088133). So, in terms of fructose and ribose transport there is no functional overlap between pGlcT and pGlcT2.

Only in the background of defective plastid maltose transport (*mex1*), which contributes by far most to the export of starch breakdown products from the plastids, the mutation of *pGlcT* has an effect on growth (shown in Cho et al., 2011 PMID: 21175634 and confirmed here). This has been interpreted as genetic evidence that pGlcT exports glucose as minor starch breakdown product from plastids. Our data further support this interpretation as we now show directly that pGlcT transports glucose, which has not been demonstrated before. Whether the compromised fructose and ribose transport capacity in *mex1 pglct* also contributes to the stronger phenotype of that double mutant compared to *mex1* is a new question that arises from our finding that pGlcT transports these sugars. However, integrating the current knowledge about the properties of pGlcT (Weber et al., 2000 PMID: 10810150; Cho et al., 2011; our data) it seems more plausible that the defect in plastid glucose transport is the main cause of the enhanced phenotype in *mex1 pglct*.

Since pGlcT2 was recently shown to also function as plastid glucose transporter (Valifard et al., 2023), we asked whether it is redundant to pGlcT (in terms of glucose transport). If it were, the triple mutant *mex1 pglct pglct2* should grow worse than any of the other mutants. However, this is not the case under the standard long-day growth conditions we have employed, which indicates that pGlcT is the main exporter for glucose from starch breakdown.

4. The title of the paper is rather perplexing, and I would suggest revising it to something more comprehensible.

REPLY: We suggest to change the title to “A plastid carbohydrate carrier is involved in ribose recycling from nucleotide catabolism and glucose export from starch degradation” but are also open to other ideas how to make the title more comprehensible.

5. The text involves pGlcT from different species, please include the Latin name abbreviations for each to distinguish them clearly, for example AtpGlcT.

REPLY: Thank you for this good suggestion. We have introduced Latin name abbreviations in many instances to improve clarity. When we speak about pGlcT in general or mean the gene/protein of several species, we have not introduced a species abbreviation. Also in figures that only show data from either Arabidopsis or bean plants, we have left out the species abbreviation, because we think that it keeps the figure legends simpler and better readable.

6. Line 31-32, to the best of my knowledge, so far, only the proton-uncoupled members of the SWEET family have been shown to transport sugars bidirectionally. However, can pGlcT, as a proton-coupled sugar transporter, also facilitate bidirectional transport?

REPLY: Given the uptake characteristics of pGlcT in the *E. coli* system, we cannot and do not claim that pGlcT acts as a proton-coupled sugar transporter. If this were the case, we would have observed a more pronounced substrate import in *E. coli* at lower external pH. As a transport optimum was observed at neutral pH (see Figure 4c and Discussion section), we rather suggest that pGlcT acts as a concentration-

driven facilitator, similar to what has been observed and proposed for its close homolog pGlcT2 (Valifard et al., 2023 PMID: 37088133). Both, pGlcT2 and pGlcT, are members of the MST transporter family and can transport sugars bidirectionally, depending on the concentration and gradient of the transported substrate across the corresponding membrane (Valifard et al., 2023 and our data). Overall, it is not uncommon for MST transporters to function as facilitators independent of a proton gradient, as has been observed for ESL1, which functions as a low-affinity facilitated diffusion transporter for glucose (Yamada et al., 2010 PMID: 19901034). Thus, facilitated sugar transport is not limited to SWEET-type transporters but is also present in selected MST-type carriers, i.e., ESL1, pGlcT2 and pGlcT.

Moreover, the presence of energized and non-energized (facilitated) sugar transport can be observed for several SUT-type transporters. Although generally described as Suc/H⁺ symporters, several transport proteins from peas and beans (PsSUF1, PsSUF4, PvSUF1), although structurally classified as SUT transporters, function as facilitators (Zhou et al., 2007 PMID: 17253986). In addition, the sucrose/H⁺ symporters PsSUT1 and PvSUT1, found in pea and bean seed coats, are thought to mediate bidirectional sucrose transport. These transporters may play a role in both, sucrose release from seed coats and, under depleted apoplastic sucrose concentrations, in sucrose efflux from sieve elements/vascular parenchyma cells (Zhou et al., 2007). Accordingly, PsSUT1 and PvSUT1 represent bona fide facilitators.

7. pGlcT is the name of plastid glucose transporter family from MST superfamily, I suggest rename this gene, like pGlcT1, 2, 3 or pGlcT-like 1.

REPLY: The name “pGlcT” for plastid glucose translocator was introduced by Weber et al. (2000, PMID: 10810150) and has been used since then in the literature. Recently, a plastid glucose transporter was found and called “pGlcT2”. With our work we clarify and extend the in vivo function of pGlcT and show that it is more than just a glucose transporter. This is not reflected in its current name, calling perhaps for a more extensive renaming.

We propose to keep the name pGlcT for now and suggest to reconsider the names for these transporters in a future review on intracellular sugar transport. Such a review could be co-authored by several prominent scientists working in this area of research, which would promote the acceptance of gene/protein renaming in this field.

8. Line 33,99,141, In vivo or in vitro needs italics.

REPLY: Journal style of Nat. Comm. is non-italics for “in vivo” and “in vitro”.

9. Line 145, what is ‘their corresponding wild types’, I do not understand, please clarify the differences between them and the wild-type Arabidopsis.

REPLY: The original sentence is: “Two T-DNA null mutants (*pglct-1*, SALK051876; *pglct-2*, SALK078684) described before³¹ and their corresponding wild types were isolated from segregating populations.” To improve the understanding, this has been rephrased to: “Two T-DNA null mutants of *At-pGlcT* (*pglct-#1*, SALK051876; *pglct-#2*, SALK078684) have been described³¹. From segregating populations of these lines we isolated the homozygous mutants and the corresponding wild types.”

Remark: This was done to have wild types that are genetically as close as possible to the null-mutants. In

our experience, this improves the metabolic analysis, as the “background noise” due to slight genetic differences is minimal.

10. To facilitate comparison, please standardize the units for ribose content and avoid using multiple notations within the same article, like ng mg⁻¹ FW, μmol g⁻¹ FW, μmol g⁻¹ DW.

REPLY: Thank you for pointing out this problem. We have changed the units in Fig. 1 and Supplementary Fig. 7 (formerly Supplementary Fig. 6) to μmol/g FW for ribose (and nmol/g FW for uridine) to be consistent with the data in Fig. 2b. These figures report metabolite concentrations after extraction of fresh material from Arabidopsis. For the metabolite measurements in bean nodules (Fig. 6a and Supplementary Fig. 8), we kept the units at μmol/g DW because the nodules were freeze dried before grinding and weighing the material for extraction. However, we have determined several times that the ratio of fresh to dry nodules is about 5. This is now mentioned in the figure legend of Fig. 6 so that the reader can quickly estimate what would have been the metabolite concentration in the fresh material.

11. To distinguish with the *pglct2* mutant, it would be preferable to rename the pGlcT mutant to *pglct#2*.

REPLY: We also noticed that the distinction between *pglct2* (mutant of gene *pGlcT2*) and *pglct-2* (second mutant allele of gene *pGlcT*) is difficult. However, these abbreviations correspond to standard Arabidopsis nomenclature. Nonetheless, we think the reviewer is right and have changed the numbering of the *pGlcT* mutant alleles to *pglct-#1* and *pglct-#2* in all cases for the sake of clarity.

12. The writing quality of this manuscript needs to be enhanced to ensure it is accessible and understandable to the broad readership of Nature Communications.

REPLY: We have made an effort to improve the writing.

Reviewer #2 (Remarks to the Author):

This manuscript aims to address an intriguing question about how ribose, derived from nucleotide degradation, is imported into plastids for recycling into metabolic processes. Researchers found that the plastid glucose transporter pGlcT is a potential player, although pGlcT transports glucose, fructose, and ribose. They also demonstrated the role of pGlcT in nodule production. However, due to the absence of a strong phenotype in the *pglct* mutant, the biological significance of pGlcT remains questionable. In other words, other transporters, rather than pGlcT, may contribute significantly.

REPLY: Our data show that pGlcT is involved in plastid ribose transport because [1] ribose accumulates in pGlcT mutants (Figs. 1 and 2); [2] ribose over-accumulates only in *pglct* background when cytosolic ribose production is boosted by external uridine feeding (Fig. 1b); [3] pGlcT transports ribose when expressed in the *E. coli* heterologous system (Fig. 4) and [4] pGlcT allows *E. coli* ribose-uptake mutants to grow better on ribose as carbon source (Fig. 3). Given this experimental evidence, we don't think that the absence of a phenotype under our growth conditions is sufficient reason to doubt the biological significance of pGlcT as ribose transporter. The absence of a growth phenotype is not uncommon for many mutants – there are numerous genes in Arabidopsis or other plants with proven biological function

whose mutants look like the wild type under standard laboratory growing conditions. However, we do agree with the reviewer that other translocators probably contribute to ribose transport across the plastid envelope. This has been argued in the results and in the discussion of the original text in lines 318 to 319 and 403 to 406 (in the revised text it is now in lines 326 to 327 and 412 to 415). Alternative ribose transport by other carriers probably becomes particularly apparent when ribose accumulates in the *pGlcT* mutant because under these conditions other transporters, whose primary *in vivo* substrate is likely not ribose, might begin to transport this sugar due to the higher substrate availability.

Major comments here:

1) The direct evidence supporting that pGlcT transports ribose, glucose, and fructose was obtained in *E. coli*. We know that proteins may not be folded the same way as in eukaryotic cells. It would be stronger if the authors could provide additional evidence supporting this in eukaryotic cells.

REPLY: Our uptake data show that pGlcT is functional in *E. coli*, hence it is very likely folded correctly. Additionally, we have now added data that show that pGlcT is incorporated into the plasma membrane of *E. coli* (new Figure 3a and Supplementary Fig. 4). Furthermore, it can be argued that a prokaryotic expression system is highly suitable to investigate transmembrane proteins of the chloroplast, which is of prokaryotic origin.

Apart from these arguments, *E. coli* is a well-accepted system for the expression of many transport proteins. We and others have exploited *E. coli* frequently for the functional expression of transport proteins from chloroplasts (Neuhaus et al., 1997 PMID: 9025303), plant mitochondria (Haferkamp et al., 2002 PMID: 12084057), the plant Endoplasmic Reticulum (Leroch et al., 2008 PMID: 18296626) and the plant plasma membrane (Rieder and Neuhaus, 2011 PMID: 21540435). Moreover, the closest homolog to pGlcT, namely pGlcT2, has also been functionally characterized in *E. coli* (Valifard et al., 2023 PMID: 37088133).

The characterization of the transport capabilities of pGlcT in *E. coli* also match well with the observations of ribose, glucose and fructose accumulation *in vivo*. Rarely do *in-vivo* and *in-vitro* data match so perfectly. Hence, there is little reason to doubt the validity of the data obtained from the heterologous *E. coli* expression system in this case, in our opinion.

2) Since a concentration of glucose 10 times higher can completely suppress ribose transport activity in substrate competition experiments (Fig. 4d), and the glucose content in leaves is approximately 10 times greater than that of ribose (Fig. 2), ribose may never have been transported via pGlcT *in vivo*.

REPLY: There is only scarce knowledge on how sugars are locally distributed in a plant cell and which local relative or absolute concentrations prevail near the transporters (see reply to query 1 from reviewer #3). So, from the *E. coli* sugar uptake studies it is impossible to conclude whether ribose is transported or is not transported *in vivo*.

However, we show that ribose accumulates in the transporter mutants, which is the gold standard for demonstrating that a substrate is transported *in vivo*. This is also supported by biological knowledge, since the only known sink for ribose in plant metabolism is ribokinase in the plastids. If the transport of ribose into the plastids is compromised, ribose accumulation would be expected. We see this accumulation in *Arabidopsis* and bean and also when we enhance cytosolic ribose production by uridine

feeding (Fig. 1b). The latter experiment does not only confirm that ribose is transported by pGlcT but it also shows that the direction of transport is from the cytosol into the plastids.

3) Quantitative data are required for Fig. 7.

REPLY: We have repeated the experiment and now provide a quantification which supports the visual impression that the Pv-pGlcT;a promoter is mainly active in infected cells of the infection zone. Please see new Fig. 7 and Supplementary Fig. 9.

4) It is unusual to depend on only one positively identified transgenic plant and to have such low expression in constitutive expression lines (Supplemental Fig. 2).

REPLY: The metabolic data in Fig. 1a shows not only one but two independent complementation lines that both fully complement the ribose accumulation phenotype. The characterization of these lines is shown in Fig. S2. One of these lines produces pGlcT without tags in *pglct* background (Fig. S2b), from the other pGlcT-YFP is produced in that background (Fig. S2a). So, additionally the data show that C-terminal YFP-tagging is possible without disturbing the function. In the time course experiment in Fig. 2b, we have only used the untagged line to reduce the enormous amounts of samples. Also here the line shows complementation of the sugar accumulation phenotypes, suggesting that the expression level of the transgene is sufficient.

We have improved the text explaining Fig. 1 to make clear that two complementation lines are shown. Previously it said: 'this molecular phenotype could be complemented by the At-pGlcT transgenes' – now it says: 'this molecular phenotype could be complemented by both the At-pGlcT and the At-pGlcT-YFP transgenes'.

Reviewer #3 (Remarks to the Author):

In this work, the authors identified and functionally characterized the first plastid ribose transporter in plants. The work is substantive and highly significant to the sugar transporter field. As the authors described in the manuscript, only a small number of plastid sugar transporters have been identified so far, which has hampered our understanding of the exchange of sugars across the inner plastid membrane. The authors first demonstrated the transport function of pGlcT for ribose, glucose and fructose in a E coli sugar-import mutant and then confirmed its in vivo function of the plastid ribose transporter in both Arabidopsis and bean, further strengthening their conclusion. I believe the authors have provided sufficient evidence to demonstrate that pGlcT is a plastid facilitator for the import of ribose from nucleotide catabolism, for the export of glucose from nocturnal starch breakdown, and for cytosol-plastid fructose exchange in vivo.

I only have two minor points for the authors to consider when revising the work.

1. As shown in Fig 4d, both glucose and fructose are competitors for ribose transport. It would be very

helpful to readers if the authors could use published concentrations of glucose and fructose (and ribose if available) in the cytosol and plastid to give readers an idea on how effective pGlcT is in transporting ribose vs. glucose and fructose in vivo.

REPLY: In the most thorough study on subcellular sugar concentrations in Arabidopsis (analyzed in the reproductive stage and harvested at mid-day) we find data only for sucrose, hexoses (sum of Fru and Glc) and raffinose (Nägele and Heyer, 2013 PMID:23488986). In plastids, the levels of sucrose range (in dependence on the ecotype analyzed) between 0.5-1.2 mM and hexoses accumulated to only about 0.2 mM. In the cytosol, sucrose ranges between 8-15 mM while hexoses are below 2 mM (Nägele and Heyer, 2013). Data on ribose levels are not available. However, given that hexoses are low in the cytosol (below 2 mM) and assuming that cellular ribose is solely located in the cytosol and plastids we predict that the measured affinity of pGlcT for ribose (about 1.2 mM) is sufficiently high to interact with this type of sugar.

Since growing conditions, time of the day, tissue (and cell type) as well as developmental age will play a role in sugar amounts and distribution, it is currently out of reach to reliably predict which sugar is when and where transported by pGlcT in vivo. This does not even consider that sugars may also be unevenly distributed within each subcellular compartment. As a first step, we would need sugar sensors in different subcellular compartments that indicate the subcellular concentrations for example by a fluorescence signal. However, our time course experiment (Fig. 2b) shows that in a 10-day-old seedling globally (in all tissues together) neither ribose nor fructose are sufficiently transported in *pglct* to keep the homeostasis of the wild type at any time. This is different for glucose, which only accumulates at night in the mutant. So apparently, occupancy of pGlcT with glucose increases during the night during starch degradation. This would mean, that it is not fully occupied with glucose during the day and would be (partially) free for other sugars.

2. The diagram presented in Fig 8 has 3 parts. I feel it could be simplified by removing Fig 8C and incorporate the transport of glucose and fructose from the cytosol to the plastid into Fig 8a and provide a couple sentences in the figure legend to indicate this possibility.

REPLY: Drawing Fig. 8a and Fig. 8c together would create a very complex figure, since metabolites and proteins involved in the processes shown in 8a and 8c are different. Since the model in Fig. 8c is more hypothetical than those in Fig. 8a and Fig. 8b, we have moved Fig. 8c into the supplementary data creating the new Supplementary Fig. 10.

REVIEWER COMMENTS

Reviewer #1 (Remarks to the Author):

The authors have improved their manuscript and answered the comments. I suggested that the current manuscript would be accepted for publication.

Reviewer #2 (Remarks to the Author):

Thank the authors for responding to my comments. But I still have some following up questions.

1. While I agree that *E. coli* has been widely used as a heterologous system to characterize transporter activity, this does not eliminate all concerns. A recent study (Andersen, 2025, PNAS, PMID: 40279393) demonstrates that different heterologous systems can yield substantial differences in kinetic parameters—up to a 30-fold difference for the same transporter. Notably, the targeting peptide was removed in this study, which could also impact substrate specificity. Therefore, I believe it is reasonable to question whether the results from *E. coli* accurately reflect the transporter's activity in *planta*.

REPLY: We agree that it is reasonable to remain critical when transporters are expressed in heterologous systems and are somehow modified (targeting peptides removed, tags added).

Reviewer #2 raises concerns that pGlcT might be misfolded in *E. coli*. This argument is made despite the fact that we can measure the activity of pGlcT in the *E. coli* system, and these activities correspond to the observed sugar accumulation in two plants within independent biological contexts. Therefore, the argument is theoretical and is not based on any inconsistencies in the given data. On the contrary, the *in vivo* data and the data from the uptake studies align very well with each other.

The theoretical argument that a protein might be misfolded can be made for any heterologous expression system - definitely also for a eukaryotic system. For example, one might speculate that partial truncation and the addition of tags disrupt protein activity, or that insertion into a non-native membrane context alters activity. These are common modifications made in any heterologous expression system.

As a eukaryotic expression system, yeast or oocytes would be available, for instance. However, neither system has plastids. For uptake studies, the transporter needs to localize to the plasma membrane, meaning that in this approach, the plastidial pGlcT would have to be modified to pass through the secretory pathway. Undoubtedly, this is in many respects not a native scenario for a plastid-localized transmembrane protein (e.g., folding at the membrane during co-translation, oxidative conditions in the ER during folding, potential unphysiological glycosylation in the Golgi, etc.). To our knowledge, there is no case in the literature where a mitochondrial or plastidial carrier was biochemically characterized after expression at the plasma membrane of a eukaryotic cell. Due to the evolutionary proximity of these organelles to prokaryotes, prokaryotic systems are suitable for this purpose—and there is a wealth of examples where *E. coli* has been used for the characterization of such transporters.

Alternatively, pGlcT could be expressed in yeast, purified, and reconstituted into liposomes. Besides the unclear membrane context in yeast for pGlcT, also in this *in vitro* system, the preservation of the native constitution of pGlcT can certainly be doubted. If successful at all, transporters frequently require renaturation and are inserted inside out.

It is also important to consider that sugars are transported by eukaryotic cells across many membrane systems, and it might be necessary to first identify suitable mutants in which pGlcT activity is measurable above the background—assuming pGlcT could be produced in sufficient quantity in these systems. The latter point is indeed critical because, despite great interest in plastid sugar transporters in the context of photosynthesis, demonstrating the activity of pGlcT has not been achieved for 25 years. Also for us, it was a significant challenge in the *E. coli* system that we could only overcome with considerable effort.

It is clear that the experimental effort that would be required to demonstrate pGlcT activity after expression in a eukaryotic cell (if successful at all) does not reasonably relate to the potential results. Most importantly, the argument from Reviewer #2 could not be conclusively refuted even by such experiments. As argued above, misfolding artifacts are also possible in a eukaryotic expression system and are no less likely than in *E. coli*.

One might argue that the additional data from the eukaryotic system would ideally confirm the data from *E. coli*, thereby providing an independent dataset that supports the activity. However, the generally accepted standard is to demonstrate the transport activity in one system. When such data match the *in vivo* data, transporter activity is considered demonstrated.

2. The authors propose that pGlcT may function bidirectionally, but no direct evidence is provided to support this claim. Does this mean that pGlcT symports ribose and glucose into the chloroplast during the day but imports ribose and exports glucose at night? If so, what is the proposed mechanism that supports such bidirectional transport with different directions of substrates?

REPLY: Our data suggest that pGlcT facilitates transport but is not energized, for example, by a proton gradient driving transport into a certain direction (see independency of transport from pH in Fig. 4c). The uridine feeding experiment (Fig. 1b), strongly suggests that transport of ribose occurs from the cytosol into the chloroplast. That glucose is probably exported by pGlcT from

the chloroplast at night is indicated by (i) the accumulation of glucose during the night in *pglct* background, (ii) the strong phenotype of *mex1 pglct* double mutants that (iii) can be rescued by supplying sucrose as shown by Cho et al. (2011, PMID: 21175634) and (iv) the ability of pGlcT to transport glucose shown here for the first time. So yes, the data suggest that pGlcT is a facilitator that equilibrates ribose, glucose and fructose over the inner plastid membrane driven merely by a concentration difference of these sugars in the two compartments. Since the metabolic sink of ribose is in the plastids but the source is in the cytosol but the main source of glucose is in the plastids, a general import function for ribose and an export function for glucose at least at night during starch breakdown are highly plausible.

Mechanistically, carriers change conformation during transport, binding a substrate on one side of the membrane and releasing it on the other side after the conformational switch. This process works in both transport directions making facilitated diffusion possible. For the human GLUT glucose transporters, for example, the process has been studied in great detail (Yan, 2017, PMID: 28756087). We suppose that pGlcT will also change conformation during transport and elucidating the mechanistic details will be a subject of future studies.

3. The lack of a strong growth phenotype generally suggests that the gene's biological significance may be limited under the tested conditions—though this does not necessarily mean the gene is functionless. Still, based on the conditions the authors specifically used, the contribution of pGlcT appears modest.

REPLY: In the context of *mex1*, the mutation of *pGlcT* has a strong visible phenotype, but not when it is mutated alone (Fig. 5). Does that mean, that plants do not need pGlcT as long as they have *Mex1*? Then I would ask why *pGlcT* is conserved in all plants. The reason must be a selective advantage that may be subtle but individuals that cannot use all glucose and ribose efficiently are apparently not as fit as the wild type and will be eliminated from a population over time in the wild. Following this argument, it must be possible to detect fitness phenotypes for mutants of all broadly conserved genes. However, experience from the laboratory shows that for many mutants macroscopic phenotypes cannot be detected, because lab conditions do not accurately reflect wild type selection pressures and are notoriously blind to long term effects on fitness.

4. According to Valifard et al. (2023, PMID: PMC10318453), pGlcT2 is not expressed in mature leaves and is only expressed in very young leaves. Given this, do the authors still believe pGlcT2 mediates glucose export from starch degradation in mature leaves? Is the pGlcT2 mutant a valid control for assessing ribose homeostasis in Fig 1? The authors should provide metabolite measurements from tissues where pGlcT2 is normally expressed in the wild type.

REPLY: Our phenotypic analysis of higher order mutants strongly suggests that pGlcT2 is not a major contributor to glucose export from starch degradation (Fig. 5 and Supplementary Fig. 5). Therefore, we state in our study (296 ff): “We conclude that pGlcT is probably the main glucose exporter downstream of plastid starch breakdown and that there is no obvious redundancy between pGlcT and pGlcT2, at least under the chosen growing conditions.” Thus, we don't believe that pGlcT2 is a major contributor to glucose export from starch degradation in mature leaves.

Phylogenetically, pGlcT2 is the closest homolog to pGlcT in *Arabidopsis*. Therefore, it was chosen as an additional control next to the various wild types that also served as controls.

Actually, it was not clear at the time these experiments were conducted, whether pGlcT2 contributes to ribose transport. Our data show that it does not, which is consistent with the transport studies reported by Valifard et al. (2023). Non-quantitative promoter-GUS data from Valifard et al. (2023) indicate that the chosen promoter fragment does not drive strong *pGlcT2* expression in mature leaves but quantitative public transcriptome and proteome data show that pGlcT2 is well expressed in all stages of leaf development (Klepikova et al, 2016, PMID: 27549386 and Mergner et al, 2020, PMID: 32188942). Additional metabolite measurements are therefore not required, in our opinion.

5. Please recheck the replicate data for uric acid in *pglct* in Fig 6a; one replicate appears to have a value even lower than any of the control replicates, which may not make sense. Is this an outlier? If excluded, the average uric acid level in *pglct* might actually be higher than in the control. How do the authors interpret this result in the context of purine metabolism?

REPLY: We checked the original data again and it is clear that uric acid was detected in that sample to the indicated level. Other signals from that sample show no abnormalities. We performed an outlier test for the uric acid signals in *pglct;a*, but no significant outliers were detected:

Descriptive Statistics

Mean: 0.08025

SD: 0.05427

of values: 4

Outlier detected?

No

Significance level:

0.05 (two-sided)

Critical value of Z:

1.4812500273

Your data

Row	Value	Z	Significant Outlier?
1	0.100	0.36390	
2	0.092	0.21650	
3	0.127	0.86138	
4	0.002	1.44178	Furthest from the rest, but not a significant outlier (P > 0.05).

6. The labeling of the complementation lines is unclear. The notation “*pglct*-#2:*pGlcT*” suggests that the *pGlcT* gene is under the control of its own promoter, but it reads as if a promoter is driving a gene in a reporter construct. A clearer format would be *pGlcT/pglct*-#2 to indicate the complementation genotype.

REPLY: We have changed the notation according to this good suggestion.

Reviewer #3 (Remarks to the Author):

The authors have addressed all the concerns I raised in the initial review.